# TOWARDS CALIBRATED DEEP CLUSTERING NETWORK

**Yuheng Jia[1,2], Jianhong Cheng[1], Hui Liu[2], Junhui Hou[3*]**
[1]School of Computer Science and Engineering, Southeast University
[2]Yam Pak Charitable Foundation School of Computing and Information Sciences,
Saint Francis University
[3]Department of Computer Science, City University of Hong Kong
{yhjia,chengjh}@seu.edu.cn, h2liu@sfu.edu.hk,
jh.hou@cityu.edu.hk

## ABSTRACT

Deep clustering has exhibited remarkable performance; however, the over-confidence problem, i.e., the estimated confidence for a sample belonging to a particular cluster greatly exceeds its actual prediction accuracy, has been over-looked in prior research. To tackle this critical issue, we *pioneer* the development of a calibrated deep clustering framework. Specifically, we propose a novel dual-head (calibration head and clustering head) deep clustering model that can effectively calibrate the estimated confidence and the actual accuracy. The calibration head adjusts the overconfident predictions of the clustering head, generating prediction confidence that matches the model learning status. Then, the clustering head dynamically selects reliable high-confidence samples estimated by the calibration head for pseudo-label self-training. Additionally, we introduce an effective network initialization strategy that enhances both training speed and network robustness. The effectiveness of the proposed calibration approach and initialization strategy are both endorsed with solid theoretical guarantees. Extensive experiments demonstrate the proposed calibrated deep clustering model not only surpasses the state-of-the-art deep clustering methods by $5\times$ on average in terms of expected calibration error, but also significantly outperforms them in terms of clustering accuracy. Code is available at https://github.com/ChengJianH/CDC.

## 1 INTRODUCTION

Clustering aims to categorize input samples into distinct groups, where samples within the same group exhibit greater similarity than those in other groups, without utilizing any ground-truth super-visory information. Recently, deep learning-based clustering methods, a.k.a. deep clustering, have showcased remarkable clustering performance owing to the powerful feature representation capability of deep neural networks (Li et al., 2021; Qian, 2023). However, training a deep clustering network is challenging as no supervisory information is available. Existing methods solve the train-ing dilemma by introducing some side tasks in the training process. For example, DEC (Xie et al., 2016) introduces the auto-encoder during the training, and (Chhabra et al., 2022) adopts a generative adversarial network.

The pseudo-labeling-based methods first extract features by self-supervised learning (like MoCo (Chen et al., 2020b)), then add a clustering head to produce the clustering prediction, and further use the model's own outputs as pseudo labels to guide the unsupervised clustering network train-ing, have become mainstream in deep clustering (Van Gansbeke et al., 2020; Niu et al., 2022; Qian, 2023). Although those methods have achieved remarkable clustering performance, they have two drawbacks. **First**, they generally use a *fixed threshold* to select the pseudo labels, which ignores the learning status of the model, i.e., in the early training stages, a higher threshold will select fewer and class-imbalanced training samples, which depress the convergence speed, while in the later training

---

*Corresponding author.

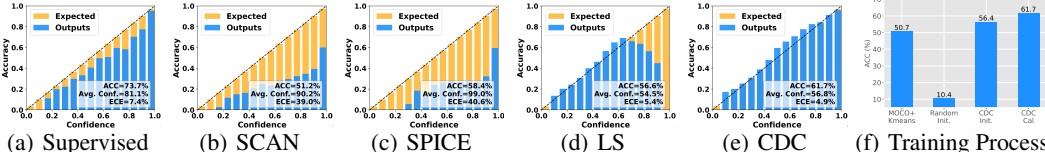

| (a) Supervised | (b) SCAN | (c) SPICE | (d) LS | (e) CDC | (f) Training Process |

Figure 1: Reliability diagrams of different methods on CIFAR-20. In the ideal case, the **confidence** of a model's output should be roughly equal to its **accuracy**, which means the confidence of the output is well-calibrated. However, the previous deep clustering models faced a severe overconfidence problem, i.e., the estimated confidence largely exceeds its real accuracy. We propose a calibrated deep clustering (CDC) model that enhances both confidence calibration and clustering performance.

stages, the fixed threshold is more likely to introduce noisy pseudo-labels due to the increasing predicted confidence. ***More critically***, those methods all implicitly assume that the output confidence of the model can reflect the true probability of a sample belonging to a certain cluster. However, those methods all face *a serious overconfidence problem*, i.e., the predicted confidence of the model's output will largely exceed the actual prediction accuracy. As shown in Fig. 1, the overconfidence problem of the state-of-the-art (SOTA) deep clustering methods like SCAN (Van Gansbeke et al., 2020) and SPICE (Niu et al., 2022) is even worse than the supervised learning model, as they rely on the pseudo-labels to train the network, and the inevitable noise in the pseudo-labels aggravates the overconfidence problem. Accurate confidence estimation is important for a machine learning model, especially in building trustworthy decision-making systems like medical diagnosis (Mimori et al., 2021), autonomous vehicles (Feng et al., 2019), and financial trading (Zhao et al., 2020), suggesting when we can trust the prediction of a learning model. However, confidence calibration has not been studied in deep clustering networks, and common calibration methods like Temperature Scaling (Guo et al., 2017a) rely on a labeled validation set, which is not accessible in deep clustering. Moreover, as shown in Fig. 1-(d), the regularization-based calibration methods like Label Smoothing (LS) (Müller et al., 2019) will over-penalize the high-reliable samples, making the prediction of reliable and unreliable samples indistinguishable. **Second**, the current methods initialize the clustering head randomly, which is unstable and will destroy the feature learned by self-supervised learning, i.e., as shown in Fig. 1-(f), the clustering accuracy (ACC) of MoCo equipped with K-means is $50.7\%$, while its performance will degrade to $10.4\%$ after random initialization.

**Our Contributions**. *Pioneering a novel approach*, we delve into the development of a meticulously calibrated deep clustering network (CDC). Our method centers on a dual-head network structure featuring a clustering head and a calibration head. The calibration head introduces a novel calibration technique that refines the prediction confidences from the clustering head across all samples, thereby providing precise confidence estimations. Simultaneously, the clustering head leverages the calibrated confidences from the calibration head to monitor class-specific learning progress and strategically select high-confidence samples for effective pseudo-labeling. This symbiotic relationship between the two heads enhances the overall performance by exchanging critical information. Additionally, we introduce a potent initialization technique for both heads. As illustrated in Fig. 1(f), our method exhibits exceptional clustering performance post the proposed initialization strategy. More importantly, the efficacy of our calibration approach and initialization strategy is underpinned by theoretical demonstrations.

Extensive evaluations on six benchmark datasets showcase the superiority of our method over state-of-the-art deep clustering methods by nearly $5\times$ on average in terms of expected calibration error (ECE) and substantial advancements in clustering accuracy.

## 2 RELATED WORK

**Deep Clustering.** Deep clustering methods can be broadly classified into two categories: *(1) Clustering based on representation learning* which first learns a representation of the data and then applies a clustering technique like K-means (Huang et al., 2014; Qi et al., 2024) and Spectral Clustering (Jia et al., 2021; Peng et al., 2021) to obtain the final clustering, *(2) Iterative deep clustering with self-supervision* which aims to learn the data representation and performs clustering simultaneously under the supervision of self-training (Xie et al., 2016; Guo et al., 2017b), self-labeling (Van Gansbeke et al., 2020; Niu et al., 2022; Qian, 2023) and contrastive information (Li et al.,

2021; Shen et al., 2021). Unlike previous methods that primarily rely on data-intrinsic features, recent methods such as SIC (Cai et al., 2023) and TAC (Li et al., 2024) use the vision-language model CLIP to mine clusters based on neighborhood consistency or pseudo labels from the textual space, which facilitates image clustering. Overall, self-labeling methods have demonstrated superior performance by selecting high-confidence instances through thresholding soft assignment probabilities and updating the entire network by minimizing the cross-entropy loss of the selected instances. However, they typically employ a fixed threshold for pseudo-label assignment, neglecting the model's learning status. How to determine a suitable threshold is still a challenging task.

**Confidence Calibration.** Accurate confidence estimation is crucial for real-world applications like self-driving cars (Feng et al., 2019) and disease diagnosis (Mimori et al., 2021), where prediction systems must not only be accurate but also indicate their likelihood of error, enhancing system interpretability. However, modern neural networks usually face the overconfidence problem, i.e., the estimated confidence by a neural network usually exceeds its real prediction accuracy (Minderer et al., 2021). To address this, calibration methods in **supervised learning** are broadly divided into two categories: *(1) Post-calibration methods* calibrate the model's output after model training. An example is Temperature Scaling (Guo et al., 2017a), which adjusts the softmax using a labeled validation set. *(2) Regularization-based methods* penalize the confidence of the model's predictions during the training process via some regularization techniques like Label Smoothing (Müller et al., 2019), which uses soft labels to reduce overfitting, and Focal Loss (Mukhoti et al., 2020), which emphasizes difficult samples and penalizes overconfident predictions. The $L_1$ Norm (Joo & Chung, 2020) is also used as an additional regularization to manage confidence levels. Recent studies have also investigated calibration under **unsupervised domain adaptation**. For instance, (Wang et al., 2020) developed a transferable calibration framework that reduces errors by accurately assessing discrepancies between source and target domains, while (Wang et al., 2023a) employs a differentiable density ratio estimator to manage uncertainties under domain shifts. These studies tackle calibration in the domain adaptation setting, where labeled source data is available.

As demonstrated in Fig. 1(b) and (c), deep clustering methods exhibit a more pronounced overconfidence issue compared to supervised learning models. This heightened problem stems from state-of-the-art deep clustering techniques relying on pseudo-labeling, where incorrect pseudo-labels exacerbate the overconfidence challenge. **Regrettably**, past research has largely overlooked this issue. Furthermore, conventional calibration techniques are unsuitable for directly addressing overconfidence in unsupervised clustering because post-calibration methods necessitate a labeled validation set for tuning hyperparameters, while regularization techniques can compromise the quality of selected pseudo-labels and, consequently, impair clustering performance. Consequently, this paper explores the development of a calibrated deep clustering neural network to simultaneously enhance clustering performance and achieve precise confidence estimation.

## 3 PROPOSED METHOD

**Overview**. As shown in Fig. 2, our model consists of a dual-head network comprising a clustering head and a calibration head. Initially, we pre-train the backbone network using MoCo-v2 (Chen et al., 2020b) and implement a feature prototype-based initialization strategy to bolster the initial discriminative capabilities of both heads (Sec. 3.3). Subsequently, we proceed to train the clustering head by leveraging high-confidence pseudo-labels through a cross-entropy loss, where the calibration head provides the confidence levels for the samples (Sec. 3.2). Concurrently, the calibration head undergoes training with a novel calibration approach to gauge and estimate sample confidence from the clustering head (Sec. 3.1). Both heads are optimized simultaneously to facilitate mutual enhancement. (Sec. 3.4)

**Notation**. Denote by $\mathcal{D}_u = \{\boldsymbol{x}_i : i \in \{1, 2, ..., N\}\}$ a training set of $N$ unlabeled samples. Weak and strong augmentation strategies from (Van Gansbeke et al., 2020) are used to obtain $\mathcal{W}(\boldsymbol{x}_i)$ and $\mathcal{A}(\boldsymbol{x}_i)$ for each sample $\boldsymbol{x}_i$. Let $f(\boldsymbol{\Theta}; \boldsymbol{x}_i)$ be the feature extractor, $g(\boldsymbol{\theta}_{clu}; \cdot)$ be the clustering head, $g(\boldsymbol{\theta}_{cal}; \cdot)$ be the calibration head, $\sigma(\cdot)$ be the softmax function, and $C$ be the predetermined classes.

### 3.1 CALIBRATION HEAD

In each training batch, we select $B$ samples from $\mathcal{D}_u$. In the remainder of this paper, all the operators are batch-wise without special mention.

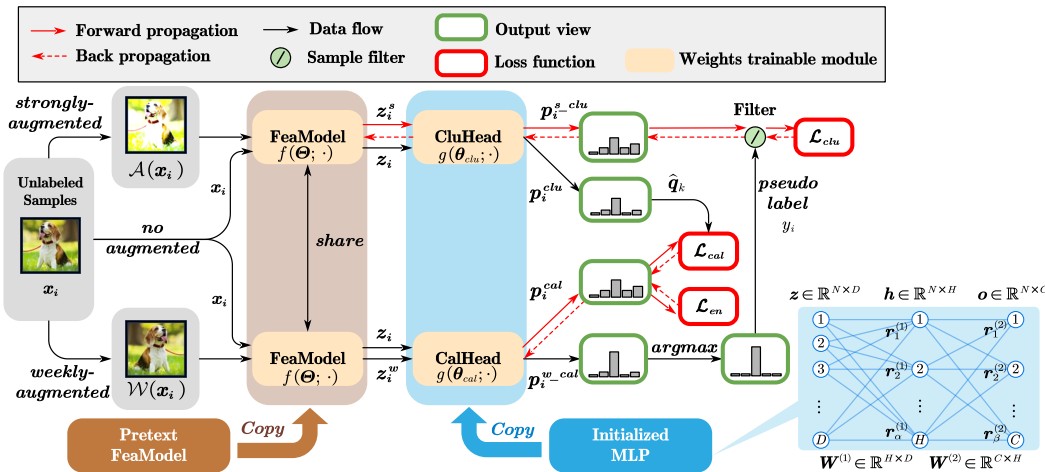

Figure 2: Illustration of the proposed CDC framework. The calibration head (CalHead) penalizes the overconfident predictions from the clustering head (CluHead). The clustering head, in turn, uses the calibrated confidence provided by the calibration head to select high-confidence samples for training. Note that the calibration head has the same structure as the clustering head.

The calibration head serves the purpose of aligning the output confidence of the model with its true accuracy. However, the well-known calibration methods are not directly applicable here: *(1) Post-calibration methods* like Temperature Scaling requires a labeled validation set to tune its hyper-parameter, which is not accessible in unsupervised clustering tasks. *(2) Regularization-based methods* like Label Smoothing (LS), Focal Loss (FL), and $L_1$ Norm (L1) penalize the prediction confidence of all the samples, which are harmful for failure detection, i.e., misclassification errors cannot be detected by filtering out low-confidence predictions. As indicated in Fig. 1-(e), although LS promotes the ECE performance, it will also reduce the high-confidence predictions.

Therefore, we selectively penalize confidence for samples in unreliable regions while preserving confidence in reliable regions. To this end, we first use the K-means algorithm to divide the embeddings matrix (before the clustering head and calibration head) into $K$ mini-clusters and record the average prediction of each mini-cluster in the clustering head as $\hat{q}_k = \frac{\sum_{\boldsymbol{x}_i \in Q_k} \boldsymbol{p}_i^{clu}}{|Q_k|}$, where $Q_k$ denotes the samples belonging to the $k$-th mini-cluster, $|Q_k|$ is the total number of samples in that mini-cluster, and the prediction of the clustering head is $\boldsymbol{p}_i^{clu} = \sigma(g(\boldsymbol{\theta}_{clu}; f(\boldsymbol{\Theta}; \boldsymbol{x}_i)))$. Then, we take the average prediction of each mini-cluster by the clustering head as the target distribution of the samples in that mini-cluster of the calibration head, i.e.,

$$\mathcal{L}_{cal} = -\frac{1}{B} \sum_k \sum_{\boldsymbol{x}_i \in Q_k} \hat{\boldsymbol{q}}_k \log \left( \boldsymbol{p}_i^{cal} \right), \tag{1}$$

where the prediction of the calibration head is $\boldsymbol{p}_i^{cal} = \sigma(g(\boldsymbol{\theta}_{cal}; f(\boldsymbol{\Theta}; \boldsymbol{x}_i)))$. The above calibration method aligns the feature representation of the samples with their output in the clustering head.

Furthermore, we add a negative entropy loss to make the predicted class distribution more uniform, avoiding all samples being divided into the same cluster:

$$\mathcal{L}_{en} = \frac{1}{C} \sum_{j=1}^{C} \boldsymbol{p}_{:,j}^{cal} \log \boldsymbol{p}_{:,j}^{cal}, \tag{2}$$

where $\boldsymbol{p}_{:,j}^{cal}$ denotes the $j$-th column of the calibration head for a batch of samples. Combining Eqs. (1) and (2), the overall loss of the calibration head is:

$$\mathcal{L} = \mathcal{L}_{cal} + w_{en}\mathcal{L}_{en}, \tag{3}$$

where we set hyperparameter $w_{en} = 1$ for simplicity.

**Theoretical Analysis.** Assuming K-means divides features into $(a + b)$ regions, where $a$ regions are *reliable regions* that do not cross clustering decision boundaries, and the set of samples in the

reliable regions are denoted as $T$, and $b$ regions are *unreliable regions* that cross the clustering decision boundaries with samples in the unreliable regions denoted as $F$. We use expected calibration error (ECE) (Naeini et al., 2015) to measure the difference between the confidence of predictions and the actual accuracy, serving as an indicator of how well-calibrated the confidence estimates are. To calculate ECE, all $N$ samples' confidence scores are grouped into $l$ bins and computed by $ECE = \sum_{i=1}^{l} \frac{|B_i|}{N} |acc(B_i) - avg.conf.(B_i)|$, where $acc(\cdot)$ and $avg.conf.(\cdot)$ means accuracy and average confidence in each bin.

**Theorem 1** (Region-aware Penalty). *Let $Conf^{clu}$ and $Conf^{cal}$ be the average confidence of the predictions before and after calibration, respectively. Then, confidence penalty occurs only in unreliable regions with $\mathbb{E}_F\left[Conf^{cal}\right] \leq \mathbb{E}_F\left[Conf^{clu}\right]$ while not in reliable regions with $\mathbb{E}_T\left[Conf^{cal}\right] = \mathbb{E}_T\left[Conf^{clu}\right]$. Proof.* See Appendix A.3.

**Theorem 2** (Improve Calibration). *The calibration errors given by the clustering head and the calibration head are denoted as $ECE^{clu}$ and $ECE^{cal}$. Under some mild conditions, we have $ECE^{cal} \leq ECE^{clu}$. Proof.* See Appendix A.4.

Theorem 1 indicates that the proposed calibration method only penalizes the confidence of the unreliable samples, which avoids the over-punishment problem on the high confidence prediction samples by the previous regularization-based calibration methods. Theorem 2 indicates the proposed calibration method can promote calibration theoretically.

**Remark**. Considering the characteristics of the calibration head outlined earlier, we implement two specific designs:

- *Stop gradient*. Given the incorporation of uncertain samples within the calibration head, we focus solely on optimizing the calibration head, steering clear of network-wide loss optimization.

- *Dual-head decoupling*. To address conflicts between calibrated and overconfident predictions within the same head, we adopt a dual-head framework to effectively separate and manage these divergent predictions.

## 3.2 CLUSTERING HEAD

Due to the lack of ground-truth information, the most crucial issue in training the clustering head is generating highly reliable pseudo labels. Previous methods (Van Gansbeke et al., 2020; Niu et al., 2022; Qian, 2023) usually adopt a *fixed* and *predefined* threshold to select samples as the pseudo labels. However, such a fashion overlooks the fact that the model's output confidence continuously increases during the training process. Specifically, in the early training strategy, the threshold is typically set too high, resulting in the selection of only a limited number of samples and slowing down the training speed. As training progresses, the confidence of all samples increases, leading to the selection of more samples as pseudo labels. Consequently, more incorrect pseudo labels are included, reducing the clustering performance. Furthermore, as previously analyzed, deep clustering networks face a significant overconfidence problem, which hinders the model's ability to perceive its *learning status*, further complicating the determination of an appropriate threshold. Additionally, previous methods utilize a single global threshold for all classes, which biases the model towards selecting more samples from easy classes (those with higher confidence) while disregarding hard classes (those with lower confidence).

To address these challenges, we use the confidence of the calibration head estimates to determine the threshold *dynamically for different classes*. This approach allows a more accurate estimation of the model's learning status and helps mitigate the discrepancies in data utilization across categories.

Specifically, we first sort all samples in class $c$ by their probability values in descending order, and then select the top $\lfloor B/C \rfloor$ samples with the highest probabilities to get $TOP(c)$, where $\lfloor \cdot \rfloor$ denotes the lower approximation operator. Then, we determine the number of samples selected as the pseudo-labels for the $c$-th cluster via

$$M(c) = \lfloor \sum_{\boldsymbol{x}_i \in TOP(c)} \boldsymbol{p}_i^{w\text{-}cal} \rfloor, \forall c = 1, 2, \cdots C, \tag{4}$$

where $\boldsymbol{p}_i^{w\text{-}cal}$ denotes the confidence of the weak augmentation of $\boldsymbol{x}_i$ estimated by the calibration head. As the calibration head is well-calibrated, the average confidence of each cluster estimated

by the calibration head can well capture the learning status of each cluster, i.e., if $M(c)$ is large (resp. small), the model is more (resp. less) confident about the prediction for the samples in the $c$-th cluster; accordingly, we should select more (resp. less) samples from this cluster as the pseudo-labels. Finally, we select the top $M_c$ samples from each cluster to constitute the final pseudo-labeled samples: $S = \{(\boldsymbol{x}_i, y_i)\}$, where $y_i = \underset{c}{\arg\max} \, \boldsymbol{p}_i^{w\text{-}cal}$ returns the cluster index producing the largest prediction as the pseudo label for $\boldsymbol{x}_i$.

After selecting the pseudo labels, we use the cross-entropy loss to optimize the parameters of the feature extractor and the clustering head:

$$\mathcal{L}_{clu} = -\frac{1}{|S|} \sum_{\boldsymbol{x}_i \in S} y_i \log \boldsymbol{p}_i^{s\text{-}clu}, \tag{5}$$

where $|S|$ denotes the total number of selected pseudo-labeled samples, $\boldsymbol{p}_i^{s\text{-}clu}$ denotes the confidence of the strong augmentation of $\boldsymbol{x}_i$ estimated by the clustering head.

### 3.3 INITIALIZATION OF CLUSTERING AND CALIBRATION HEADS

After pre-training the backbone using MoCo-v2, a clustering head, typically a multilayer perceptron (MLP), is added to generate clustering predictions. Figure 1-(f) illustrates that the MoCo-v2 pre-trained backbone yields a quite discriminative representations, for example, applying K-means to MoCo-v2 pre-trained features on the CIFAR-20 dataset achieves a clustering accuracy of about 50.7%. In contrast, a randomly initialized clustering head on this backbone only reaches an accuracy of 10.4%. This indicates that the discriminative power of the pre-trained backbone does not automatically transfer to the clustering head. Consequently, the common practice of random initialization can result in low-quality pseudo-labels and poor clustering performance.

Therefore, we propose a feature prototype-based initialization strategy for the clustering head (also the same approach for the calibration head) to extend the discriminative capabilities of the pre-trained backbone. As detailed in the lower right corner of Fig. 2, we employ a three-layer MLP for the clustering head. This MLP consists of $D$ input units, $H$ hidden units, and $C$ output units. The variables include $\boldsymbol{z} \in \mathbb{R}^{N \times D}$ (input), $\boldsymbol{h} \in \mathbb{R}^{N \times H}$ (hidden), $\boldsymbol{o} \in \mathbb{R}^{N \times C}$ (output), and two linear layer weights $\boldsymbol{W}^{(1)} \in \mathbb{R}^{H \times D}$ and $\boldsymbol{W}^{(2)} \in \mathbb{R}^{C \times H}$. To initialize the first linear layer $\boldsymbol{W}^{(1)}$, we perform K-means clustering on the input variable $\boldsymbol{z}$, and split it into $H$ mini-clusters. Then we use each clustering center (prototype) to initialize $\boldsymbol{W}^{(1)}$, i.e.,

$$\boldsymbol{W}^{(1)} = \texttt{Kmeans}_H(\boldsymbol{z}), \tag{6}$$

where $\texttt{Kmeans}_H(\boldsymbol{z})$ returns $H$ prototypes of $\boldsymbol{z}$ by the K-means clustering. Note that we set the bias of the linear layer to 0. By the above setting, one neuron of the output will denote a prototype, and this neuron will yield large values (resp. small values) for samples belonging to (resp. not belonging) the mini-cluster related to that prototype, marking it has the discriminative ability transferred from the pre-trained backbone. Besides, the MLP also embeds a BN layer and a ReLU layer. For the second layer weight $\boldsymbol{W}^{(2)}$, we apply the same strategy by initializing with K-means on $\boldsymbol{h}$, resulting in $\boldsymbol{W}^{(2)} = \texttt{Kmeans}_C(\boldsymbol{h})$. After the initialization with feature prototypes, we further orthogonalize the weights to improve the discriminative ability of the network.

**Proposition 1.** *Assuming $\boldsymbol{z}^*$ as the prototype obtained by applying K-means to the features $\boldsymbol{z}$, when the linear layer weights $\boldsymbol{W} = \boldsymbol{z}^*$, the cluster assignment of the output $\boldsymbol{W}\boldsymbol{z}$ aligns well with $\boldsymbol{z}^*$.*

*Proof.* See Appendix A.1                                                                                    □

Proposition 1 demonstrates that the proposed strategy can effectively propagate the discriminability from the feature space to the output space, thereby outperforming random initialization.

### 3.4 JOINT TRAINING AND FINAL PREDICTION

**Joint Optimization.** After the pre-training by MoCo-v2, and the initialization of the clustering head and the calibration head, we optimize the clustering head and feature extractor by minimizing Eq. 5

and optimize the calibration head by minimizing Eq. 3. The overall training process of our model is summarized in Algorithm 1.

**Final Prediction**. The clustering head achieves clustering results similar to that of the calibration head but with a worse ECE. Therefore, we use the calibration head to make the final prediction of $i$-th sample $\boldsymbol{x}_i$, i.e., $\boldsymbol{p}_i^{cal} = \sigma\left(g\left(\boldsymbol{\theta}_{cal}; f\left(\boldsymbol{\Theta}; \boldsymbol{x}_i\right)\right)\right)$.

---

**Algorithm 1:** The training process of **C**alibrated **D**eep **C**lustering

---

**Data:** Training dataset $\mathcal{D}_u = \{\boldsymbol{x}_i : i \in \{1, 2, ..., N\}\}$
**Input:** $C$ predefined clusters, $K$ mini-clusters; feature extractor $f(\boldsymbol{\Theta}; \cdot)$, the clustering head $g(\boldsymbol{\theta_{clu}}; \cdot)$, the calibration head $g(\boldsymbol{\theta_{cal}}; \cdot)$, and softmax function $\sigma(\cdot)$.
**Initialization:**
Learning feature extractor $\boldsymbol{\Theta}$ via MoCo-v2;
Initializing MLPs $\boldsymbol{\theta}_{clu}$ and $\boldsymbol{\theta}_{cal}$ by the proposed initialization approach;
**for** $epoch \leftarrow 1$ **to** *EPOCHS* **do**
    **for** $b \leftarrow 1$ **to** $\lfloor N/B \rfloor$ **do**
        Establish batch dataset $\mathcal{D}_b \subseteq \mathcal{D}_u$;
        $\boldsymbol{p}_i^{w\_cal} \xleftarrow{no\ grad.} \sigma\left(g\left(\boldsymbol{\theta}_{cal}; f\left(\boldsymbol{\Theta}; \mathcal{W}(\boldsymbol{x}_i)\right)\right)\right)$;
        Establish pseudo-labeled samples $S$ by Eqs. 4;
        $\boldsymbol{z_i} \xleftarrow{no\ grad.} f\left(\boldsymbol{\Theta}; \boldsymbol{x}_i\right)$
        $\boldsymbol{p}_i^{clu} \xleftarrow{no\ grad.} \sigma\left(g\left(\boldsymbol{\theta}_{clu}; f\left(\boldsymbol{\Theta}; \boldsymbol{x}_i\right)\right)\right)$;
        Employ K-means on $\boldsymbol{z}$ to get partition $Q_k$;
        Calculate mini-clusters target distribution $\hat{\boldsymbol{q}}_k$;
        **for** $sub\_iter \leftarrow 1$ **to** $\lfloor B/B_s \rfloor$ **do**
            Establish sub-batch dataset $\mathcal{D}_{sub} \subseteq \mathcal{D}_b$;
            $\boldsymbol{p}_{sub}^{s\_clu} \leftarrow \sigma\left(g\left(\boldsymbol{\theta}_{clu}; f\left(\boldsymbol{\Theta}; \mathcal{A}(\boldsymbol{x}_{sub})\right)\right)\right)$;
            Calculate $\mathcal{L}_{clu}$ by Eq. 5, update $\boldsymbol{\Theta}$ and $\boldsymbol{\theta}_{clu}$;
            $\boldsymbol{p}_{sub}^{cal} \leftarrow \sigma\left(g\left(\boldsymbol{\theta}_{cal}; f\left(\boldsymbol{\Theta}; \boldsymbol{x}_{sub}\right)\right)\right)$;
            Calculate $\mathcal{L}_{cal}$ and $\mathcal{L}_{en}$ by Eqs. 1-2, update $\boldsymbol{\theta}_{cal}$;
        **end**
    **end**
**end**

---

## 4 EXPERIMENTS

### 4.1 EXPERIMENT SETTINGS

**Datasets and Backbones**. We conducted experiments on six widely used benchmark datasets, CIFAR-10, CIFAR-20 (Krizhevsky et al., 2009), STL-10 (Coates et al., 2011), ImageNet-10, ImageNet-Dogs (Chang et al., 2017), and Tiny-ImageNet (Le & Yang, 2015). Similar to (Huang et al., 2022) and (Niu et al., 2022), we used ResNet-34 and MLP (512d-BN (Ioffe & Szegedy, 2015)-ReLU (Nair & Hinton, 2010)-$C$d) for all experiments.

**Implementation Details**. The comparison methods can be divided into two types. *(1) Clustering based on representation learning* which can achieve clustering by using K-means (Lloyd, 1982): MoCo-v2 (Chen et al., 2020b), SimSiam (Chen et al., 2020a), BYOL (Grill et al., 2020), DMICC (Li et al., 2023), ProPos (Huang et al., 2022) and CoNR (Yu et al., 2024). *(2) Iterative deep clustering with self-supervision*: DivClust (Metaxas et al., 2023), CC (Li et al., 2021), TCC (Shen et al., 2021), TCL (Li et al., 2022), SeCu (Qian, 2023), SCAN (Van Gansbeke et al., 2020), and SPICE (Niu et al., 2022). More details are shown in B.1.

For CDC, we trained the model for 100 epochs using the Adam optimizer with a learning rate of 5e-5 for the encoder and 1e-4 for the MLP. The learning rate of the encoder on CIFAR-20 and Tiny-ImageNet was adjusted to 1e-5 for better learning of noisy pseudo labels. We set B=1,000 for CIFAR-10, CIFAR-20, and STL-10, B=500 for ImageNet-10 and ImageNet-Dogs, B=5,000 for Tiny-ImageNet. We set K=500 for CIFAR-10, K=40 for CIFAR-20 and ImageNet-Dogs, K=150 for STL-10 and ImageNet-10, and K=1,000 for Tiny-ImageNet. For simplicity, the predictions of the clustering and calibration heads are denoted as **CDC-Clu** and **CDC-Cal**, respectively.

Table 1: The clustering performance ACC, ARI (%) and calibration error ECE (%) of various deep clustering methods trained on six image benchmarks. The best and second-best results are highlighted in **bold** and underlined, respectively. ↑ (↓) means the higher (resp. lower), the better.

| Method | CIFAR-10 | | | CIFAR-20 | | | STL-10 | | | ImageNet-10 | | | ImageNet-Dogs | | | Tiny-ImageNet | | |
|---|---|---|---|---|---|---|---|---|---|---|---|---|---|---|---|---|---|---|
| | ACC↑ | ARI↑ | ECE↓ | ACC↑ | ARI↑ | ECE↓ | ACC↑ | ARI↑ | ECE↓ | ACC↑ | ARI↑ | ECE↓ | ACC↑ | ARI↑ | ECE↓ | ACC↑ | ARI↑ | ECE↓ |
| K-means | 22.9 | 4.9 | N/A | 13.0 | 2.8 | N/A | 19.2 | 6.1 | N/A | 24.1 | 5.7 | N/A | 10.5 | 2.0 | N/A | 2.5 | 0.5 | N/A |
| MoCo-v2 | 82.9 | 64.9 | N/A | 50.7 | 26.2 | N/A | 68.8 | 45.5 | N/A | 56.7 | 30.9 | N/A | 62.8 | 48.1 | N/A | 25.2 | 11.0 | N/A |
| Simsiam | 70.7 | 53.1 | N/A | 33.0 | 16.2 | N/A | 49.4 | 34.9 | N/A | 78.4 | 68.8 | N/A | 44.2 | 27.3 | N/A | 19.0 | 8.4 | N/A |
| BYOL | 57.0 | 47.6 | N/A | 34.7 | 21.2 | N/A | 56.3 | 38.6 | N/A | 71.5 | 54.1 | N/A | 58.2 | 44.2 | N/A | 11.2 | 4.6 | N/A |
| DMICC | 82.8 | 69.0 | N/A | 46.8 | 29.1 | N/A | 80.0 | 62.5 | N/A | 96.2 | 91.6 | N/A | 58.7 | 43.8 | N/A | - | - | - |
| ProPos | 94.3 | 88.4 | N/A | 61.4 | 45.1 | N/A | 86.7 | 73.7 | N/A | 96.2 | 91.8 | N/A | 77.5 | 67.5 | N/A | 29.4 | 17.9 | N/A |
| CoNR | 93.2 | 86.1 | N/A | 60.4 | 44.3 | N/A | 92.6 | 84.6 | N/A | 96.4 | 92.2 | N/A | **79.4** | 66.7 | N/A | 30.8 | 18.4 | N/A |
| DivClust | 81.9 | 68.1 | - | 43.7 | 28.3 | - | - | - | - | 93.6 | 87.8 | - | 52.9 | 37.6 | - | - | - | - |
| CC | 85.2 | 72.8 | 6.2 | 42.4 | 28.4 | 29.7 | 80.0 | 67.7 | 11.9 | 90.6 | 85.3 | 8.1 | 69.6 | 56.0 | 19.3 | 12.1 | 5.7 | **3.2** |
| TCC | 90.6 | 73.3 | - | 49.1 | 31.2 | - | 81.4 | 68.9 | - | 89.7 | 82.5 | - | 59.5 | 41.7 | - | - | - | - |
| TCL | 88.7 | 78.0 | - | 53.1 | 35.7 | - | 86.8 | 75.7 | - | 89.5 | 83.7 | - | 64.4 | 51.6 | - | - | - | - |
| SeCu-Size | 90.0 | 81.5 | 8.1 | 52.9 | 38.4 | 13.1 | 80.2 | 63.1 | 9.9 | - | - | - | - | - | - | - | - | - |
| SeCu | 92.6 | 85.4 | 4.9 | 52.7 | 39.7 | 41.8 | 83.6 | 69.3 | 6.5 | - | - | - | - | - | - | - | - | - |
| SCAN-2 | 84.1 | 74.1 | 10.9 | 50.0 | 34.7 | 37.1 | 87.0 | 75.6 | 7.4 | 95.1 | 89.4 | 2.7 | 63.3 | 49.6 | 26.4 | 27.6 | 15.3 | 27.4 |
| SCAN-3 | 90.3 | 80.8 | 6.7 | 51.2 | 35.6 | 39.0 | 91.4 | 82.5 | 6.6 | 97.0 | 93.6 | 1.5 | 72.2 | 58.7 | 19.5 | 25.8 | 13.4 | 48.8 |
| SPICE-2 | 84.4 | 70.9 | 15.4 | 47.6 | 30.3 | 52.3 | 89.6 | 79.2 | 10.1 | 92.1 | 83.6 | 7.8 | 64.6 | 47.7 | 35.3 | 30.5 | 16.3 | 48.5 |
| SPICE-3 | 91.5 | 83.4 | 7.8 | 58.4 | 42.2 | 40.6 | 93.0 | 85.5 | 6.3 | 95.9 | 91.2 | 4.1 | 67.5 | 52.6 | 32.5 | 29.1 | 14.7 | N/A |
| CDC-Clu (Ours) | 94.9 | **89.4** | 1.4 | **61.9** | **46.7** | 28.0 | **93.1** | **85.8** | 4.8 | 97.2 | 94.0 | 1.8 | 79.3 | **70.3** | 17.1 | **34.0** | **20.0** | 37.8 |
| CDC-Cal (Ours) | **94.9** | **89.5** | **1.1** | 61.7 | 46.6 | **4.9** | 93.0 | 85.6 | **0.9** | **97.3** | **94.1** | **0.8** | 79.2 | 70.0 | **7.7** | 33.9 | 19.9 | 11.0 |
| Supervised | 89.7 | 78.9 | 4.0 | 71.7 | 50.2 | 11.0 | 80.4 | 62.2 | 10.0 | 99.2 | 98.3 | 0.9 | 93.1 | 85.7 | 0.9 | 47.7 | 24.3 | 5.1 |
| +MoCo-v2 | 94.1 | 87.5 | 2.4 | 83.2 | 68.4 | 6.7 | 90.5 | 80.7 | 3.5 | 99.9 | 99.8 | 0.4 | 99.5 | 99.0 | 0.9 | 53.8 | 30.9 | 8.4 |

**Evaluation Metrics**. We used three common clustering metrics: clustering Accuracy (ACC) (Li & Ding, 2006), Normalized Mutual Information (NMI) (Strehl & Ghosh, 2002), and Adjusted Rand Index (ARI) (Hubert & Arabie, 1985) to evaluate the clustering performance, and used the expected calibration error (ECE) (Naeini et al., 2015) to evaluate the calibration performance. Moreover, we applied the failure rejection ability metrics AUROC, AURC, and FPR95 (Zhu et al., 2022) to evaluate the separation quality of low-confidence and high-confidence predictions.

## 4.2 MAIN RESULTS

**Superior Clustering Ability.** As shown in Tab. 1, applying K-means to the pre-trained features significantly outperforms directly applying K-means to the raw features, demonstrating the importance of deep representation learning in clustering. Along this line, ProPos and CoNR have continuously improved the clustering performance but cannot provide confidence prediction. For self-labeling-based methods, CDC-Cal outperforms DivClust, CC, SeCu, SCAN, and SPICE on almost all the datasets, validating the importance of dynamic threshold setting. On CIFAR-10 and STL-10, CDC-Cal with the same pre-trained model (MoCo-v2) increases accuracy by 0.8% and 2.5% than supervised learning. On large-scale datasets, CDC-Cal has achieved improvements of 1.4%, 11.7%, and 4.8% on ImageNet-10, ImageNet-Dogs, and Tiny-ImageNet. Meanwhile, the results of the clustering head CDC-Clu are close to those of the calibration head CDC-Cal. In summary, our method ranks first in 11 out of 12 cases in all the clustering metrics, suggesting its superior clustering ability.

**Excellent Calibration Performance.** The calibration error of CDC-Cal is far superior to that of CDC-Clu and all the compared methods. CC does not apply the pseudo-labeling strategy, so its ECE is relatively low. This explains that overfitting to the wrong pseudo-labels is a reason for the high calibration error. SPICE-2 uses dual-softmax losses to encourage sharper predictions, which leads to more severe overconfidence than SCAN, while our calibration head significantly reduces the calibration error on CIFAR-10 (15.4%→1.1%), CIFAR-20 (52.3%→4.9%), and STL-10 (7.8%→0.8%) compared to SPICE-2. Fig. 1 provides a visual comparison of the ECE on CIFAR-20, where our method achieves well-aligned predictions and better clustering performance simultaneously.

**Competitive Failure Rejection Ability.** Fig. 3 compares different regularization-based calibration methods. CDC-Cal improves AUROC by 8.6%, decreases AURC by 10.6%, and decreases FPR95 by 6.5% compared to the second-best method, demonstrating its effective error rejection capability. The second line of Fig. 3 further demonstrates our method can better separate correct and mis-

classified samples compared with the regularization-based calibration methods, which is beneficial for selecting reliable pseudo-labels for feedback into the network. Although Focal Loss (FL) may produce a lower ECE than our method, its clustering ACC is much worse. Besides, our method outperforms FL on all the failure rejection metrics (AUROC, AURC, and FPR95), suggesting our method can better separate the correct prediction and the wrong prediction.

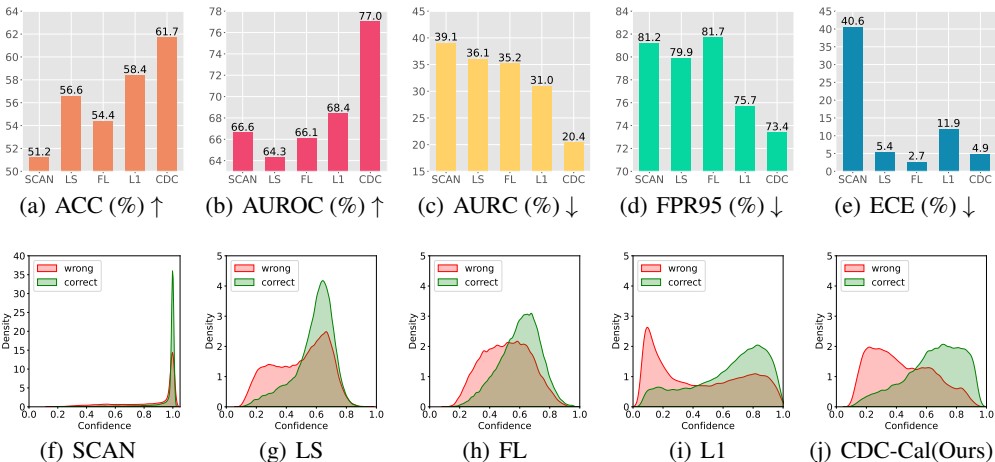

Figure 3: The failure rejection ability comparison on CIFAR-20. The second row shows the confidence distribution of correct and misclassified samples, demonstrating that our method has a stronger ability to separate failure predictions.

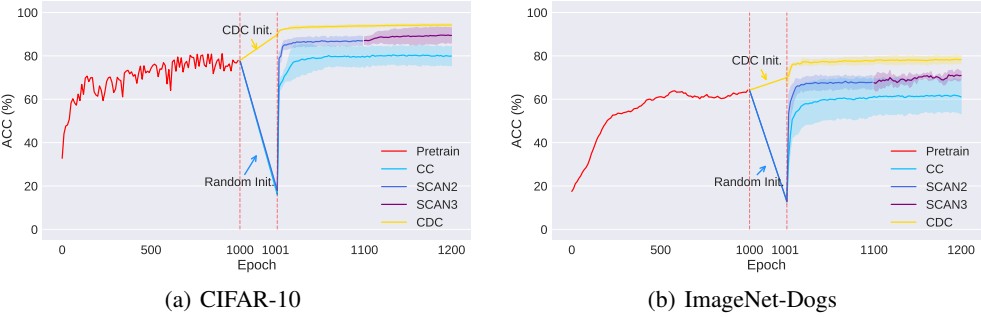

Figure 4: The training process on CIFAR-10 and ImageNet-Dogs. CDC-Cal has (i) fewer training stages, (ii) better initialization strategy, and (iii) more stable performance improvement.

## 4.3 ABLATION STUDY

**I. Initialization.** Fig. 4 shows the benefits of our initialization strategy in improving training speed and stability. Our method can transform the discriminative ability of the pre-trained features to the clustering head directly. Tab. 2-I quantitatively demonstrates that prototype-based initialization leads to a notable enhancement in clustering performance. Specifically, when we exclude the initialization stage from our method (w/o Init.+CDC), the clustering performance experiences a significant decline, showing the efficacy of the initialization strategy.

**II. Confidence-Aware Selection.** Tab. 2-II shows that if we fix the threshold at certain values, the clustering performance will drop significantly, suggesting the effectiveness of the proposed confidence-aware thresholding strategy.

**III. Single-head Setting.** Tab. 2-III shows in a single-head setting, the network employs its own overconfident predictions for confidence-aware sample selection, which may lead to the selection of numerous incorrect pseudo-labels, resulting in performance degradation including both the clustering performance and the calibration performance.

Table 2: The ablation results of the proposed model.

| Type | Settings | CIFAR-10 | | | | CIFAR-20 | | | | STL-10 | | | |
|------|----------|----------|------|------|------|----------|------|------|------|--------|------|------|------|
| | | ACC↑ | NMI↑ | ARI↑ | ECE↓ | ACC↑ | NMI↑ | ARI↑ | ECE↓ | ACC↑ | NMI↑ | ARI↑ | ECE↓ |
| **I** | After Randomly Init. | 19.1 | 7.6 | 3.1 | 8.5 | 10.4 | 5.7 | 1.0 | 4.9 | 19.2 | 6.0 | 2.4 | 8.5 |
| | After Proposed Init. | 87.2 | 79.8 | 76.1 | 1.0 | 56.4 | 56.9 | 41.2 | 5.2 | 89.8 | 80.9 | 79.3 | 2.7 |
| | w/o Init.+CDC | 89.4 | 86.5 | 83.5 | 3.3 | 44.4 | 52.3 | 31.0 | 11.9 | 73.3 | 70.0 | 60.6 | 17.9 |
| **II** | Fixed Thre. (0.99) | 80.6 | 69.8 | 65.8 | 6.5 | 54.9 | 55.9 | 37.1 | 15.1 | 89.6 | 81.0 | 79.3 | 1.0 |
| | Fixed Thre. (0.95) | 91.9 | 85.2 | 83.9 | 3.5 | 50.8 | 49.2 | 30.5 | 12.6 | 91.8 | 84.0 | 83.3 | 1.1 |
| | Fixed Thre. (0.90) | 92.7 | 86.5 | 85.3 | 3.2 | 43.3 | 43.2 | 27.3 | 4.1 | 93.0 | 86.1 | 85.7 | 1.0 |
| | Fixed Thre. (0.80) | 93.6 | 87.5 | 86.9 | 1.7 | 49.9 | 50.5 | 33.6 | 3.9 | 93.0 | 86.1 | 85.7 | 2.0 |
| | CDC-Cal (Ours) | 94.9 | 89.3 | 89.5 | 1.1 | 61.7 | 60.9 | 46.6 | 4.9 | 93.0 | 85.8 | 85.6 | 0.9 |
| **III** | Single-head (Clu) | 93.9 | 88.0 | 87.5 | 2.3 | 59.7 | 61.3 | 45.3 | 31.6 | 92.6 | 85.3 | 84.9 | 5.1 |
| **IV** | Single-head (Clu+Cal) | 94.8 | 89.0 | 89.1 | 1.8 | 57.8 | 58.7 | 43.1 | 21.2 | 93.0 | 85.7 | 85.5 | 3.0 |
| **V** | Cal (w/o Stop Gradient) | 93.0 | 86.0 | 85.7 | 2.0 | 49.6 | 52.1 | 34.2 | 12.4 | 86.7 | 76.3 | 63.9 | 2.5 |

**IV. Dual-head Decoupling.** Tab. 2-IV shows that combining the conflicting losses of encouraging overconfidence and penalizing confidence in a single head (Clu+Cal) will decrease both the calibration performance and the clustering performance especially on CIFAR-20.

**V. Stop Gradient for the Calibrating Head.** If we do not stop the gradient for the calibration head, the clustering performance will degrade significantly, especially on CIFAR-20 and STL-10. This is because the calibration loss relies on unreliable regions to provide penalties, which will include unreliable information in the feature extractor if we do not stop the gradient.

**VI. Mini-cluster Number K.** The value of K represents the number of regions and directly influences the confidence penalty. The choice of K depends on the complexity of the dataset. For challenging datasets with over-confident predictions, a lower K value is needed to increase the confidence penalty, with the extreme case being K = 1, where all samples approach a uniform distribution, resulting in the maximum penalty. For simpler datasets, a higher K value is necessary to reduce the confidence penalty. If K equals the number of samples, each sample is independent, and the penalty is zero. Fig. 5 shows how this trend affects confidence. On CIFAR-20, variations of K by ±20% lead to ACC changes of ±1.3% and ECE changes of ±0.5%, which is acceptable. More ablation studies are shown in the Appendix B.5.

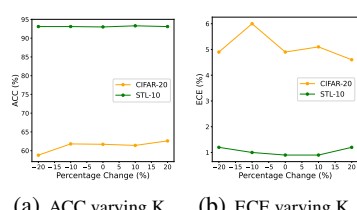

(a) ACC varying K    (b) ECE varying K

Figure 5: The ACC and ECE are robust to varying K.

## 5 CONCLUSION

In this paper, we have introduced a novel calibrated deep clustering model that outperforms SOTA deep clustering models in terms of both clustering performance and calibration performance. Notably, our model is the first deep clustering method capable of calibrating the output confidence. In particular, the expected calibration error of our is 5 times better than certain compared methods. To achieve these results, we proposed a dual-head clustering network consisting of a clustering head and a calibration head. The clustering head utilizes the calibrated confidence estimated by the calibration head to select highly reliable pseudo-labels. The calibration head, on the other hand, calibrates the output confidence of all samples using a novel regularization loss. Additionally, we introduced an effective initialization strategy for both the clustering head and the calibration head, significantly promoting training stability. Through our calibrated deep clustering neural network, the output confidence of the model is well aligned with its true accuracy. Moreover, as we do not need any label to train the network, the above benefit can be gained quite easily.

ACKNOWLEDGMENTS

This work was supported in part by the National Natural Science Foundation of China under Grants U24A20322 and 62422118, and in part by Hong Kong UGC under Grants UGC/FDS11/E02/22 and UGC/FDS11/E03/24. This work is also supported by the Big Data Computing Center of Southeast University.

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

## A THEORETICAL PROOF

### A.1 PROOF OF PROPOSITION 1

**Proposition 1** Assuming $z^*$ as the prototype obtained by applying K-means to the features $z$, when the linear layer weights $W = z^*$, the cluster assignment of the output $Wz$ aligns well with $z^*$.

*proof.*

Without loss of generality, we assume the features $\mathbf{z}$ is normalized. When conducting K-means on feature $z$ to derive prototypes $z^*$, we need to solve the following optimization problem,

$$
\begin{aligned}
min_{z^*}||z - z^*||^2 &= min_{z^*} - 2z^*z + ||z||^2 + ||z^*||^2 \\
&= min_{z^*} - z^*z = max_{z^*}z^*z \xmapsto{W=z^*} max_W Wz
\end{aligned}
\tag{7}
$$

The above process demonstrates that the optimal clustering assignment in the feature space is also applicable in the output space, indicating consistency between the feature space and the output space.

### A.2 INTUITIVE EXPLANATION

Recalling the setup in Sec. 3.1, we use K-means to divide the features into $(a + b)$ regions, where $a$ regions are **Reliable Regions** that do not cross clustering decision boundaries and $b$ regions are **Unreliable Regions** that cross the clustering decision boundaries.

In the reliable region (the left half of Fig. 6), the mean predicted value for four samples within this region is $[0.97, 0.03]$, which is consistent with the average maximum confidence of $0.97$ for each sample. In the unreliable region (the right half of Fig. 6), the mean value voted by four samples within this region is $[0.45, 0.55]$, while the average maximum confidence of these samples is $0.70$. This inconsistency will be the source of confidence penalty by our method.

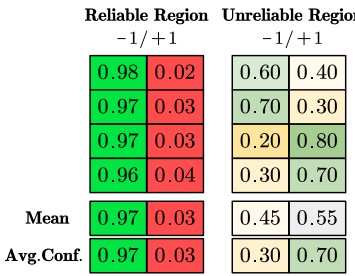

Figure 6: Calibration Example

Therefore, by identifying regions with similar features in the feature space, our method can maintain predictions in reliable regions, and penalize predictions in unreliable regions. Accordingly, high confidence will be kept and better failure rejection ability will can be achieved.

### A.3 PROOF OF THEOREM 1

**Theorem 1 (Region-aware Penalty)** The average confidence of the predictions before and after calibration are denoted as $Conf^{clu}$ and $Conf^{cal}$. Then, confidence penalty occurs only in unreliable regions with $\mathbb{E}_F [Conf^{cal}] \leq \mathbb{E}_F [Conf^{clu}]$ while not in reliable regions with $\mathbb{E}_T [Conf^{cal}] = \mathbb{E}_T [Conf^{clu}]$.

*proof.*

Considering a binary classification problem like (Zhang et al., 2022), let's assume that the features of the data $(z, y) \in \mathbb{R}^m \times \{-1, 1\}$ are generated from a mixture of two Gaussian distributions, where each class corresponds to one Gaussian distribution. Then, the following conditional distribution holds: $z|y \sim N(y\theta^*, \sigma^2 I)$. Assuming the classifier is $C(z) = sgn(\hat{\theta}^T z)$, where $\hat{\theta} = \sum_{i=1}^n \frac{z_i y_i}{n}$. The confidence vector is denoted as $[p_{+1}(z), p_{-1}(z)]^T$, with $p_c(z) = \frac{1}{\exp(-2c\hat{\theta}^T z_i / \sigma^2) + 1}$, where $c \in \{-1, +1\}$.

In reliable region, we have

$$
\mathbb{E}_T [Conf^{cal}] = \mathbb{E}_{z \in T} [p_c^{cal}(z)] = \mathbb{E}_{z \in T} [\bar{p}_c^{clu}(z)] = \mathbb{E}_{z \in T} [p_c^{clu}(z)] = \mathbb{E}_T [Conf^{clu}].
$$

In unreliable region, we have

$$
\begin{aligned}
\mathbb{E}_{F}\left[Conf^{cal}\right] &= \mathbb{I}\left(\mathbb{E}_{z\in F}\left[p_{+1}^{cal}\left(z\right)\right]\geq\frac{1}{2}\right)\mathbb{E}_{z\in F}\left(p_{+1}^{cal}\left(z\right)\right)+\mathbb{I}\left(\mathbb{E}_{z\in F}\left[p_{-1}^{cal}\left(z\right)\right]\geq\frac{1}{2}\right)\mathbb{E}_{z\in F}\left(p_{-1}^{cal}\left(z\right)\right)\\
&= \mathbb{I}\left(\mathbb{E}_{z\in F}\left[p_{+1}^{cal}\left(z\right)\right]\geq\frac{1}{2}\right)\left(\mathbb{E}_{z\in F,\hat{\boldsymbol{\theta}}^{T}z<0}\left(p_{+1}^{cal}\left(z\right)\right)+\mathbb{E}_{z\in F,\hat{\boldsymbol{\theta}}^{T}z\geq0}\left(p_{+1}^{cal}\left(z\right)\right)\right)\\
&\quad+\mathbb{I}\left(\mathbb{E}_{z\in F}\left[p_{-1}^{cal}\left(z\right)\right]\geq\frac{1}{2}\right)\left(\mathbb{E}_{z\in F,\hat{\boldsymbol{\theta}}^{T}z<0}\left(p_{-1}^{cal}\left(z\right)\right)+\mathbb{E}_{z\in F,\hat{\boldsymbol{\theta}}^{T}z\geq0}\left(p_{-1}^{cal}\left(z\right)\right)\right)\\
&= \mathbb{I}\left(\mathbb{E}_{z\in F}\left[p_{+1}^{cal}\left(z\right)\right]\geq\frac{1}{2}\right)\left(\mathbb{E}_{z\in F,\hat{\boldsymbol{\theta}}^{T}z<0}\left[p_{-1}^{cal}\left(z\right)-\left(p_{-1}^{cal}\left(z\right)-p_{+1}^{cal}\left(z\right)\right)\right]\right)\\
&\quad+\mathbb{I}\left(\mathbb{E}_{z\in F}\left[p_{+1}^{cal}\left(z\right)\right]\geq\frac{1}{2}\right)\mathbb{E}_{z\in F,\hat{\boldsymbol{\theta}}^{T}z\geq0}\left(p_{+1}^{cal}\left(z\right)\right)\\
&\quad+\mathbb{I}\left(\mathbb{E}_{z\in F}\left[p_{-1}^{cal}\left(z\right)\right]\geq\frac{1}{2}\right)\mathbb{E}_{z\in F,\hat{\boldsymbol{\theta}}^{T}z<0}\left(p_{-1}^{cal}\left(z\right)\right)\\
&\quad+\mathbb{I}\left(\mathbb{E}_{z\in F}\left[p_{-1}^{cal}\left(z\right)\right]\geq\frac{1}{2}\right)\left(\mathbb{E}_{z\in F,\hat{\boldsymbol{\theta}}^{T}z\geq0}\left[p_{+1}^{cal}\left(z\right)-\left(p_{+1}^{cal}\left(z\right)-p_{-1}^{cal}\left(z\right)\right)\right]\right)\\
&= \left[\mathbb{I}\left(\mathbb{E}_{z\in F}\left[p_{+1}^{cal}\left(z\right)\right]\geq\frac{1}{2}\right)+\mathbb{I}\left(\mathbb{E}_{z\in F}\left[p_{-1}^{cal}\left(z\right)\right]\geq\frac{1}{2}\right)\right]\mathbb{E}_{z\in F,\hat{\boldsymbol{\theta}}^{T}z<0}\left(p_{-1}^{cal}\left(z\right)\right)\\
&\quad+\left[\mathbb{I}\left(\mathbb{E}_{z\in F}\left[p_{+1}^{cal}\left(z\right)\right]\geq\frac{1}{2}\right)+\mathbb{I}\left(\mathbb{E}_{z\in F}\left[p_{-1}^{cal}\left(z\right)\right]\geq\frac{1}{2}\right)\right]\mathbb{E}_{z\in F,\hat{\boldsymbol{\theta}}^{T}z\geq0}\left(p_{+1}^{cal}\left(z\right)\right)\\
&\quad-\mathbb{I}\left(\mathbb{E}_{z\in F}\left[p_{+1}^{cal}\left(z\right)\right]\geq\frac{1}{2}\right)\mathbb{E}_{z\in F,\hat{\boldsymbol{\theta}}^{T}z<0}\left[p_{-1}^{cal}\left(z\right)-p_{+1}^{cal}\left(z\right)\right]\;(\mathbf{I})\\
&\quad-\mathbb{I}\left(\mathbb{E}_{z\in F}\left[p_{-1}^{cal}\left(z\right)\right]\geq\frac{1}{2}\right)\mathbb{E}_{z\in F,\hat{\boldsymbol{\theta}}^{T}z\geq0}\left[p_{+1}^{cal}\left(z\right)-p_{-1}^{cal}\left(z\right)\right]\;(\mathbf{II})\\
&= \mathbb{E}_{z\in F,\hat{\boldsymbol{\theta}}^{T}z<0}\left(p_{-1}^{cal}\left(z\right)\right)+\mathbb{E}_{z\in F,\hat{\boldsymbol{\theta}}^{T}z\geq0}\left(p_{+1}^{cal}\left(z\right)\right)-(\mathbf{I})-(\mathbf{II})\\
&= \mathbb{E}_{F}\left[Conf^{clu}\right]-(\mathbf{I})-(\mathbf{II})
\end{aligned}
$$

As when $\hat{\boldsymbol{\theta}}^{T}z<0$ , we have $p_{-1}^{cal}\left(z\right)-p_{+1}^{cal}\left(z\right)=p_{-1}^{clu}\left(z\right)-p_{+1}^{clu}\left(z\right)\geq0$.

While when $\hat{\boldsymbol{\theta}}^{T}z\geq0$ , we have $p_{+1}^{cal}\left(z\right)-p_{-1}^{cal}\left(z\right)=p_{+1}^{clu}\left(z\right)-p_{-1}^{clu}\left(z\right)\geq0$.

Hence, $\mathbb{E}_{F}\left[Conf^{clu}\right]-\mathbb{E}_{F}\left[Conf^{cal}\right]=(\mathbf{I})+(\mathbf{II})\geq0$ and we can conclude that our method can penalty over-confidence predictions only in the uncertain regions.

### A.4 PROOF OF THEOREM 2

**Theorem 2 (Improve Calibration)** The calibration errors given by the clustering head and the calibration head are denoted as $ECE^{clu}$ and $ECE^{cal}$. Under some mild conditions, we have $ECE^{cal}\leq ECE^{clu}$.

*proof.*

First, ECE is defined as:

$$
ECE = \mathbb{E}_{\boldsymbol{v}=\hat{\boldsymbol{\theta}}^{T}z}\left|ACC\left(\boldsymbol{v}\right)-p_{c}\left(z\right)\right|
$$

In reliable region, we have

$$
ECE_{T}^{clu}-ECE_{T}^{cal}=\mathbb{E}_{\boldsymbol{v}_{a}=\hat{\boldsymbol{\theta}}^{T}z,z\in T}\left[\left|ACC^{clu}\left(\boldsymbol{v}_{a}\right)-p_{c}^{clu}\left(z\right)\right|-\left|ACC^{cal}\left(\boldsymbol{v}_{a}\right)-p_{c}^{cal}\left(z\right)\right|\right].
$$

As the two heads have the same clustering assignments, we have

$$\mathbb{E}_{T}\left[ACC^{clu}\right] = \mathbb{E}_{T}\left[ACC^{cal}\right],\ \mathbb{E}_{T}\left[Conf^{clu}\right] = \mathbb{E}_{T}\left[Conf^{cal}\right],$$

then

$$ECE_{T}^{clu} - ECE_{T}^{cal} = 0.$$

In unreliable region, we have

$$ECE_{F}^{clu} - ECE_{F}^{cal} = \mathbb{E}_{\boldsymbol{v}_b = \hat{\boldsymbol{\theta}}^T \boldsymbol{z}, \boldsymbol{z} \in F}\left[\left|ACC^{clu}\left(\boldsymbol{v}_b\right) - p_c^{clu}\left(\boldsymbol{z}\right)\right| - \left|ACC^{cal}\left(\boldsymbol{v}_b\right) - p_c^{cal}\left(\boldsymbol{z}\right)\right|\right].$$

As the calibration head uses the predictions of the clustering head, resulting in similar clustering assignment outcomes for both, thus we have

$$ACC^{clu}\left(\boldsymbol{v}_b\right) \approx ACC^{cal}\left(\boldsymbol{v}_b\right) = ACC.$$

$(i)$ If $p_c^{clu}\left(\boldsymbol{z}\right) > ACC$, $p_c^{cal}\left(\boldsymbol{z}\right) > ACC$, the model faces overconfidence, we have

$$ECE_{F}^{clu} - ECE_{F}^{cal} = \mathbb{E}_{\boldsymbol{z} \in F}\left[p_c^{clu}\left(\boldsymbol{z}\right) - p_c^{cal}\left(\boldsymbol{z}\right)\right]$$
$$= (\mathbf{I}) + (\mathbf{II}) \geq 0$$

$(ii)$ If $p_c^{clu}\left(\boldsymbol{z}\right) > ACC$, $p_c^{cal}\left(\boldsymbol{z}\right) < ACC$, and $\left(\mathbb{E}_{\boldsymbol{z} \in F}\left[p_c^{clu}\left(\boldsymbol{z}\right)\right] + \mathbb{E}_{\boldsymbol{z} \in F}\left[p_c^{cal}\left(\boldsymbol{z}\right)\right]\right) \geq \mathbb{E}_{F}ACC$
holds, we have

$$ECE_{F}^{clu} - ECE_{F}^{cal} = \mathbb{E}_{\boldsymbol{z} \in F}\left[p_c^{clu}\left(\boldsymbol{z}\right)\right] + \mathbb{E}_{\boldsymbol{z} \in F}\left[p_c^{cal}\left(\boldsymbol{z}\right)\right] - 2\mathbb{E}_{F}ACC \geq 0.$$

$(iii)$ If $p_c^{clu}\left(\boldsymbol{z}\right) < ACC$, $p_c^{cal}\left(\boldsymbol{z}\right) < ACC$, the model faces underconfidence. We do not consider this situation as deep networks are more prone to overconfidence.

$(iv)$ If $p_c^{clu}\left(\boldsymbol{z}\right) < ACC$, $p_c^{cal}\left(\boldsymbol{z}\right) > ACC$, due to $p_c^{clu}\left(\boldsymbol{z}\right) \geq p_c^{cal}\left(\boldsymbol{z}\right)$ always holds in Theorem 1, there is no solution.

In summary, we can get

$$ECE^{cal} \leq ECE^{clu}.$$

## B  EXPERIMENTAL DETAILS AND MORE RESULTS

### B.1  IMPLEMENTATION DETAILS

**Datasets**. As shown in Tab. 3, we conducted experiments on six widely used benchmark datasets, CIFAR-10 (Krizhevsky et al., 2009), CIFAR-20 (Krizhevsky et al., 2009), STL-10 (Coates et al., 2011), ImageNet-10 (Chang et al., 2017), ImageNet-Dogs (Chang et al., 2017), and Tiny-ImageNet (Le & Yang, 2015). Similar to (Van Gansbeke et al., 2020), we used 20 superclasses of the CIFAR-100 dataset to construct the CIFAR-20, and 10, 15, and 200 subclasses

Table 3: A summary of the datasets

| Dataset | #Samples | #Classes | Image Size |
|---|---|---|---|
| CIFAR-10 | 60,000 | 10 | 32×32 |
| CIFAR-20 | 60,000 | 20 | 32×32 |
| STL-10 | 13,000 | 10 | 96×96 |
| ImageNet-10 | 13,000 | 10 | 224×224 |
| ImageNet-Dogs | 19,500 | 15 | 224×224 |
| Tiny-ImageNet | 100,000 | 200 | 64×64 |

of the ImageNet-1k (Deng et al., 2009) dataset to extract the ImageNet-10, ImageNet-Dogs, and Tiny-ImageNet. Meanwhile, we extended STL-10 by 100,000 relevant unlabeled data during the pretext training and removed them afterwards. We used the dataset from ImageNet-10, ImageNet-Dogs, and Tiny-ImageNet datasets for both training and testing, while the rest datasets were trained and tested on the merged datasets.

**Backbones**. To facilitate training on small datasets such as CIFAR-10 and CIFAR-20, we followed (Huang et al., 2022) and modified the first convolution layer's kernel of the ResNet-34 with a kernel

size of 3 × 3, padding of 2 and stride of 1, and removed the first max-pooling layer. Moreover, All experiments are conducted on an NVIDIA RTX 3090 GPU.

**Representation Learning.** For MoCo-v2, SimSiam and BYOL, according to (Van Gansbeke et al., 2020), (Chen et al., 2020a), (Chen et al., 2020b) and (Grill et al., 2020), we apply the same augmentation strategy for all datasets including a random ResizedCrop with an image size reported in Tab. 3 and a random HorizontalFlip, followed by a random ColorJitter and a random Grayscale. Moreover, we use GaussianBlur except for CIFAR-10 and CIFAR-20 due to the size of image. We adopt the ResNet-34 as backbone and a 2-layer MLP head (hidden layer 4096-d, with BN and ReLU, output layer 256-d) as projector. Besides, BYOL use another similar 2-layer MLP head as predictor. For opitimizer, MoCo-v2 and SimSiam adopt SGD optimizer, while BYOL adopt LARS (You et al., 2017) optimizer. we train all these methods over 1,000 epochs with base learning rate 0.5 and batch size 256 on the datasets shown in Tab. 3. We update the learning rate with a cosine decay learning rate schedule (Loshchilov & Hutter, 2016) and start with a warmup (Goyal et al., 2017) for 50 epochs. For the momentum hyperparameter $\tau$, we set 0.99 and 0.996 for MoCo-v2 and BYOL. For other hyperparameters, we set the temperature and weight decay as 0.2 and 0.0001 for all methods, and queue size 32768 for MoCo-v2.

**Comparison Method.** *(1) Clustering based on representation learning.* The parameter settings for MoCo-v2, SimSiam, and BYOL have been outlined previously. For ProPos, due to the difference of backbone, we re-implemented the results on Tiny-ImageNet dataset using ResNet-34. For other datasets, since the backbone, data augmentation and dataset partition scheme are the same as ours, we cite the results of (Huang et al., 2022). For DMICC (Li et al., 2023), and CoNR (Yu et al., 2024), we directly cited the results of the paper. *(2) Iterative deep clustering with self-supervision.* For methods requiring a pre-training stage (CC, SCAN, SPICE), we selected the MoCo-v2 model. During the formal training statge, under a consistent data augmentation and dataset partitioning scheme, we used ResNet-34 to re-implement CC, SCAN and SeCu. For CC, we used a smaller initial learning rate of 5e-5 to fine-tune the model, and the rest of the parameters were consistent with (Li et al., 2021). For SCAN and SeCu, we used the same hyperparameters as the (Van Gansbeke et al., 2020) and (Qian, 2023). For SPICE, except for SPICE-3 used WideResNet-28-2 for Cifar10, WRN-28-8 for CIFAR-20, and WRN-37-2 for STL-10, the rest of the experiments used ResNet-34, and these results are from the official model library[1]. Since SPICE-3 has not released their model on the Tiny-ImageNet dataset, we directly cited the results and did not test the model's ECE. For DivClust (Metaxas et al., 2023), TCC (Shen et al., 2021) and TCL (Li et al., 2022), we directly cited the results of the paper.

**Supervised Baseline.** We trained on the training datasets of CIFAR-10, CIFAR-20, STL10 and Tiny-ImageNet, and evaluated the results on the test or validation datasets. Due to a lack of the designated test datasets on ImageNet-10 and ImageNet-Dogs, we reported the results on the training datasets. We used Adam optimizer with weight decay of 0.0001 for all the experiments, and the batch size was set to 256. We adopted cross entropy on strong augmented samples to train a baseline model for 500 epochs with an initial learning rate of 0.001, and fine-tuned the MoCo-v2 model 50 epochs with an initial learning rate of 0.0001 to obtain a better baseline.

**Sample Selection Strategy**. For our method, the clustering head selects only high-confident samples for training, and the calibration network utilizes all samples during the training process. The above strategy aims to obtain a more accurate estimation of the overall confidence and enhance the model's robustness.

**Augmentation Strategy**. For the clustering head of our method, we employed weakly augmented samples to generate pseudo-labels and used the strongly augmented samples as the distribution to be learned. For the calibration head, no augmentation is included to eliminate any representation differences caused by sample transformations.

## B.2 RELIABILITY DIAGRAMS

The reliability diagrams for all models trained on CIFAR-20 are shown in Fig. 7. Our proposed method demonstrates lower calibration error when the overall average confidence and clustering accuracy are comparable.

---

[1]https://github.com/niuchuangnn/SPICE

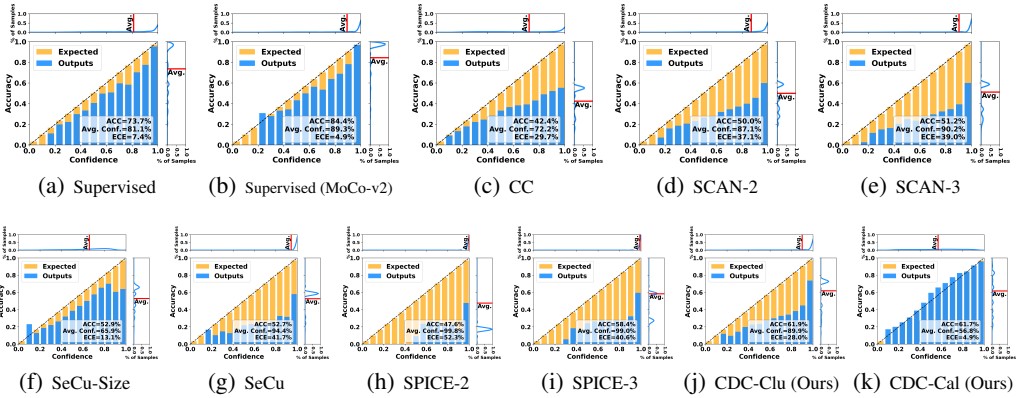

Figure 7: Reliability diagrams on CIFAR-20.

## B.3 Clustering Performance: NMI

In addition to the ACC and ARI results shown in Sec. 4.2, our method also achieved 5 out of 6 best results on NMI (Tab. 4).

Table 4: The clustering performance NMI on six image benchmarks. The best results are shown in **bold**, while the second best results are underlined.

| Method | CIFAR-10 | CIFAR-20 | STL-10 | ImageNet-10 | ImageNet-Dogs | Tiny-ImageNet |
|---|---|---|---|---|---|---|
| K-means | 8.7 | 8.4 | 12.5 | 11.9 | 5.5 | 6.5 |
| MoCo-v2 | 77.7 | 56.4 | 69.3 | 54.1 | 62.8 | 43.0 |
| Simsiam | 67.7 | 32.6 | 61.8 | 81.7 | 45.9 | 36.4 |
| BYOL | 65.5 | 43.9 | 67.0 | 70.3 | 63.4 | 29.6 |
| DMICC | 74.0 | 45.2 | 68.9 | 91.7 | 58.1 | - |
| ProPos | 88.6 | 60.6 | 75.8 | 90.8 | 73.7 | 46.0 |
| CoNR | 86.7 | 60.4 | 85.2 | 91.1 | 74.4 | 46.2 |
| DivClust | 72.4 | 44.0 | - | 89.1 | 51.6 | - |
| CC | 76.9 | 47.1 | 72.7 | 88.5 | 65.4 | 32.5 |
| TCC | 79.0 | 47.9 | 73.2 | 84.8 | 55.4 | - |
| TCL | 81.9 | 52.9 | 79.9 | 87.5 | 62.3 | - |
| SeCu-Size | 79.3 | 51.6 | 69.4 | - | - | - |
| SeCu | 86.1 | 55.1 | 73.3 | - | - | - |
| SCAN-2 | 79.3 | 52.2 | 78.8 | 88.9 | 60.1 | 43.1 |
| SCAN-3 | 82.5 | 51.2 | 83.6 | 92.5 | 67.4 | 40.7 |
| SPICE-2 | 73.5 | 44.3 | 80.4 | 82.8 | 56.9 | 44.9 |
| SPICE-3 | 85.4 | 57.6 | **86.1** | 90.2 | 62.6 | 42.7 |
| CDC-Clu (Ours) | 89.3 | 60.6 | 86.0 | 93.1 | **76.7** | 47.2 |
| CDC-Cal (Ours) | **89.3** | **60.9** | 85.8 | **93.2** | 76.5 | **47.5** |
| Supervised | 78.8 | 59.8 | 67.3 | 97.7 | 86.6 | 59.3 |
| +MoCo-v2 | 86.9 | 72.9 | 81.8 | 99.7 | 98.7 | 64.0 |

## B.4 Difference between Label Smoothing (LS) and CDC

LS reduces confidence for all samples uniformly, while our method adopts a region-aware penalty that only reduces the confidence for unreliable regions while keeping the high confidence for reliable regions. As shown in Fig. 8, LS lacks high-confidence samples (e.g., confidence > 0.9) and shows

overlapping peaks for correct and incorrect predictions with different hyper-parameters. This indicates that LS, regardless of the hyperparameter adjustments, tends to over-penalize high-confidence samples, leading to diminished model performance.

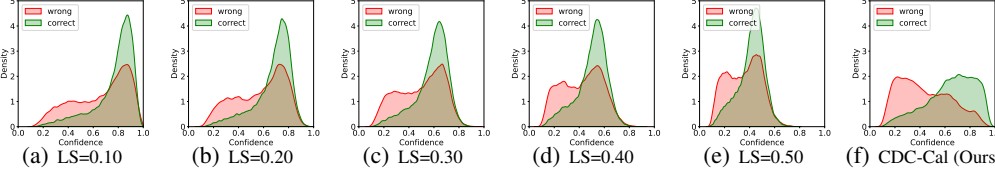

Figure 8: (a)(b)(c)(d)(e) Label Smoothing (LS) shows worse separation between correct and incorrect predictions, regardless of hyperparameter adjustments from 0.10 to 0.50. (f) CDC-Cal (Ours).

## B.5 ABLATION STUDY

**Batch size and Number of mini-clusters (B, K)**. As demonstrated in Tab. 5, our model exhibits considerable robustness to variations within a reasonable range for B and K, especially on challenging datasets such as CIFAR-20. Variations of K by ±20% and B by ±50% only result in the changes in overall accuracy of ±1.4%.

- **Insight for B**. We constrain GPU memory usage to within 9GB. For datasets with smaller image size, larger batch sizes are utilized (e.g., B=1000 for CIFAR-10 and CIFAR-20, which have image sizes of 32×32). Conversely, smaller batch sizes are employed for datasets with larger image sizes (e.g., B=500 for ImageNet-10 and ImageNet-Dogs, which have image sizes of 224×224).

- **Insight for K**. K is tuned empirically according to the complexity of the dataset. For challenging datasets with over-confident predictions, a lower K value is needed to increase the confidence penalty, with an empirical ratio of $B/K \leq 5$. Conversely, for simpler datasets, a higher K value is necessary to reduce the confidence penalty, supported by an empirical ratio of $B/K \geq 10$.

Table 5: Influence of (B, K) on clustering accuracy ACC (%) and expected calibration error ECE (%) across STL-10 and CIFAR-20 datasets. "Std." indicates the standard deviation of the data in this row.

| Dataset | #B | 500 | | | 1000 | | | 1500 | | | Std. |
|---------|-----|------|------|------|------|------|------|------|------|------|------|
| | #K | -20% | 0% | +20% | -20% | 0% | +20% | -20% | 0% | +20% | |
| STL-10 | ACC | 93.0 | 93.1 | 93.2 | 93.1 | 93.0 | 93.1 | 92.8 | 92.8 | 93.1 | 0.1 |
| | ECE | 1.1 | 1.2 | 1.6 | 1.2 | 0.9 | 1.2 | 1.2 | 1.0 | 0.8 | 0.2 |
| CIFAR-20 | ACC | 58.1 | 61.0 | 60.2 | 58.8 | 61.7 | 62.6 | 59.1 | 60.0 | 59.2 | 1.4 |
| | ECE | 6.9 | 7.6 | 3.6 | 4.9 | 4.9 | 4.6 | 4.4 | 1.4 | 3.9 | 1.7 |

Table 6: Influence of Hidden layer size and weight $w_{en}$ on clustering accuracy ACC (%) and expected calibration error ECE (%) across STL-10 and CIFAR-20 datasets.

| Dataset | #Hidden | 256 | 384 | 512 | 640 | 768 | #$w_{en}$ | 0.0 | 0.1 | 0.5 | 1.0 | 5.0 | 10.0 |
|---------|---------|------|------|------|------|------|-----------|------|------|------|------|------|------|
| STL-10 | ACC↑ | **93.2** | 93.1 | 93.0 | 92.6 | 92.2 | ACC↑ | 92.9 | 93.0 | 92.8 | 93.0 | 93.1 | **93.2** |
| | ECE↓ | 1.7 | 1.1 | 0.9 | **0.6** | 1.0 | ECE↓ | 1.1 | 1.0 | 1.3 | **0.9** | 0.9 | 1.0 |
| CIFAR-20 | ACC↑ | 61.7 | 60.6 | **61.7** | 58.3 | 57.1 | ACC↑ | 57.0 | 57.7 | 58.8 | **61.7** | 58.2 | 56.9 |
| | ECE↓ | 3.7 | 4.1 | 4.9 | 3.5 | **2.5** | ECE↓ | 2.4 | 2.7 | 1.8 | 4.9 | 2.7 | **1.7** |

**Hidden Layer Size (H)**. The choice of 512-dim is commonly used in MLP, and provides a good balance between initialization complexity and performance, as Tab. 6 shows empirically.

**Loss Weight** ($w_{en}$). As shown in Tab. 6, our method is robust to $w_{en}$, so we fix it at $w_{en} = 1$ for simplicity in the experiments. Moreover, removing the negative entropy loss $L_{en}$ will degrade the clustering performance, especially on CIFAR-20.

**Advantage of the Proposed Calibration Loss over the Other Regularization-based Calibration Losses**. We provide more detailed reliability diagrams on CIFAR-20, where the calibration head is applied with different regularization losses. Although these methods can reduce calibration errors, their clustering performance does not significantly improve. This is related to the penalty imposed on high-confidence samples by these methods: over-penalizing high-confidence samples, which is evident in the reliability diagram in Fig. 9. Since samples with similar features are more likely to have consistent outputs, our method imposes a smaller penalty on high-confidence samples than on low-confidence samples, which effectively maintains the high confidence predictions of reliable samples.

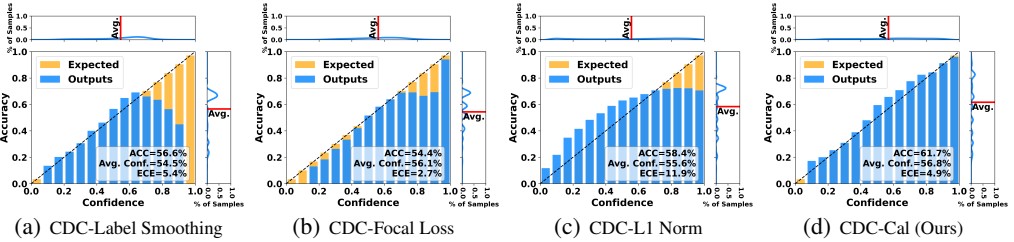

(a) CDC-Label Smoothing     (b) CDC-Focal Loss     (c) CDC-L1 Norm     (d) CDC-Cal (Ours)

Figure 9: Reliability diagrams on CIFAR-20 with the calibration head applied different regularization losses. In the high-confidence region, all compared calibration losses face a decline in clustering accuracy.

## B.6 TIME AND SPACE COMPLEXITY

Table 7: Running time and memory requirements of SCAN and CDC. All experiments run on one 3090 GPU and ResNet-34.

| | | CIFAR-10 | CIFAR-20 | STL-10 | ImageNet-10 | ImageNet-Dogs | Tiny-ImageNet |
|---|---|---|---|---|---|---|---|
| Million Parameters of CDC | | 21.8 | 21.8 | 21.8 | 21.8 | 21.8 | 22.0 |
| Running Time (Hour) | SCAN2 (100 eps) +SCAN3 (200 eps) | 4.7 | 4.6 | 1.3 | 4.0 | 5.5 | 10.3 |
| | CDC (100 eps) | 8.7 (+4.1) | 2.3 (-2.3) | 1.3 (-0.1) | 6.0 (+2.0) | 5.8 (+0.3) | 5.6 (-4.7) |
| GPU Memory (MB) | SCAN2 | 5789 (bs=256) | 5798 (bs=256) | 9237 (bs=256) | 13489 (bs=256) | 18907 (bs=256) | 6480 (bs=256) |
| | SCAN3 | 5478 (bs=1000) | 5479 (bs=1000) | 7598 (bs=1000) | 10764 (bs=256) | 12608 (bs=256) | 7414 (bs=1000) |
| | CDC | 5620 (bs=1000) | 4849 (bs=1000) | 5822 (bs=1000) | 8396 (bs=500) | 8396 (bs=500) | 7138 (bs=5000) |

As shown in Tab. 7, our dual-head network does not significantly increase the number of parameters compared to the original ResNet-34 (21.80 million parameters). Regarding GPU memory, we use a combination of batch and sub-batch strategies (see Algorithm 1 in 3.4), leading to larger batch sizes and lower memory usage compared to competing methods under the same optimizer.

For running time, we condensed the two-stage process (totaling 300 epochs) used in SCAN into a single stage (only 100 epochs). We speed up K-Means with K-Means++ initialization and the PyTorch implementation as in ProPos (Huang et al., 2022). Therefore, we save 2.3 hours on CIFAR-20 and 4.7 hours on Tiny-ImageNet. Although training time increases on CIFAR-10 and ImageNet-10 due to the large number of mini-clusters, the potential benefits in terms of enhanced failure rejection capability and improved clustering performance often outweigh these costs.

# C    Further Discussion

## C.1    Insights from Semi-Supervised Learning (SSL)

### C.1.1    Enhancing Self-Training with Moderately Confident Samples

Based on (Tang & Jia, 2022), we explore more moderately confident samples via Gradient Synchronization Filter (GSF) and Prototype Proximity Filter (PPF) to boost learning, which is conceptually aligned with our approach to pseudo-labeling. From the Tab. 8, we find that GSF and PPF can only bring marginal improvements in performance. The reason is the highly overlap in sample selection between the CDC's strategy and those selected by GSF and PPF, the expansion of selected samples is less than 5%.

Nevertheless, related techniques hold potential to further enhance our model, which we intend to explore thoroughly in the future.

Table 8: Clustering performance (ACC, NMI, ARI %) on STL-10 and CIFAR-20.

| Method | STL-10 | | | CIFAR-20 | | |
|---|---|---|---|---|---|---|
| | ACC↑ | NMI↑ | ARI↑ | ACC↑ | NMI↑ | ARI↑ |
| CDC | 93.03 | 85.85 | 85.57 | 61.66 | 60.94 | 46.57 |
| CDC+GSF | 92.98 | 85.77 | 85.48 | 61.92 | 61.20 | **46.79** |
| CDC+PPF | **93.03** | **85.79** | **85.57** | **62.12** | **61.30** | 46.57 |

### C.1.2    Enhancing Self-Training with Dynamic Thresholding in SSL

We explore the application of the thresholding strategies from FlexMatch (Zhang et al., 2021) and FreeMatch (Wang et al., 2023b) in our CDC framework by two approaches (**Apply Directly** and **Integration**).

**Apply Directly**. Consistent with CDC, after pre-training with MoCo-v2 and performing prototype-based initialization, we utilize pseudo-label learning, evaluating the learning state based on the model's output. As shown in the first two rows of Tab. 9, CDC outperforms FlexMatch and FreeMatch on both datasets. The reasons are as follows. FlexMatch and FreeMatch rely on the prediction confidence estimated by a well-calibrated model, which is reasonable in semi-supervised learning (SSL) scenarios where some labels are known. In deep clustering, the model updates thresholds based on over-confident predictions, which can lead to the selection of more noisy labels, causing performance degradation-this effect is more pronounced in challenging datasets like CIFAR-20. Specifically, selecting 80.2% of the overall samples corresponds to an accuracy of 54.9% in FlexMatch, selecting 90.8% of the overall samples also corresponds to an accuracy of 48.8% in FreeMatch, whereas selecting 55.2% of the overall samples corresponds to an accuracy of 61.7% in CDC.

**Integration.** Consistent with the previous settings, only the threshold selection strategy of CDC is replaced by FlexMatch and FreeMatch. As shown in the last two rows of Tab. 9, CDC outperforms FlexMatch and FreeMatch on both datasets. The reasons are as follows. Under the predictions of a well-calibrated model, FlexMatch requires a dataset-specific global threshold. When the global threshold for STL-10 and CIFAR-20 is set to 0.95 as recommended by (Zhang et al., 2021), it results in significantly different sample selection scales (91.2% in STL-10 and 3.2% in CIFAR-20), affecting performance improvements. FreeMatch is more robust, similar to CDC, with both methods being free from manually setting thresholds. The difference between FreeMatch and CDC on CIFAR-20 lies in the handling of head classes (classes with a large number of selected samples): 1) FreeMatch uses local threshold adjustments to modify the overall threshold (the average of the

maximum probability values for all samples), with local thresholds after maxNorm ranging from 0 to 1. Thus, the thresholds for head classes are less than the overall threshold. Accordingly, FreeMatch will introduce more samples for these classes (and more wrong pseudo-labels). 2) CDC directly selects reliable samples based on calibrated probabilities for each class, ensuring that the thresholds for head classes are not reduced by the overall threshold.

Table 9: Clustering performance ACC, NMI, ARI(%) and calibration error ECE (%) of Semi-Supervised Learning methods applied directly to self-labeling (first two rows) and integrated into CDC (last two rows) on STL-10 and CIFAR-20. Sel. represents the proportion of selected samples.

| Method | STL-10 | | | | | CIFAR-20 | | | | |
|---|---|---|---|---|---|---|---|---|---|---|
| | ACC ↑ | NMI↑ | ARI↑ | ECE↓ | Sel. (%) | ACC ↑ | NMI↑ | ARI↑ | ECE↓ | Sel. (%) |
| FlexMatch (Zhang et al., 2021) | 92.6 | 85.6 | 84.8 | 5.8 | 90.6 | 54.9 | 55.6 | 39.2 | 37.4 | 80.2 |
| FreeMatch (Wang et al., 2023b) | 93.0 | **85.9** | 85.6 | 5.1 | 88.3 | 48.8 | 54.9 | 16.8 | 44.0 | 90.8 |
| CDC-Cal | **93.0** | 85.8 | **85.6** | 0.9 | 89.4 | **61.7** | **60.9** | **46.6** | 4.9 | 55.2 |
| CDC+FlexMatch | 92.3 | 84.6 | 84.0 | 2.0 | 91.2 | 57.2 | 59.4 | 42.1 | 6.5 | 3.2 |
| CDC+FreeMatch | 92.9 | 85.6 | 85.3 | **0.7** | 81.9 | 57.8 | 59.1 | 43.6 | **1.4** | 64.3 |

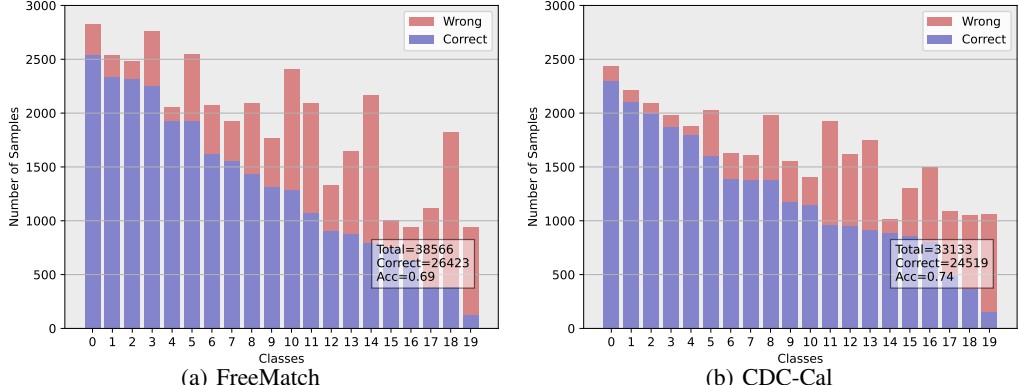

(a) FreeMatch         (b) CDC-Cal

Figure 10: The distribution of selected high-confidence samples on each class (sorted by the number of correctly predicted samples per class) after 100 epochs of training on CIFAR-20. "Total" and "Correct" represent the number of selected and correctly predicted high-confidence samples, respectively, while "ACC" represents the accuracy of selected samples. Among the top 5 classes with the highest number of correct predictions, FreeMatch selectes 12,671 samples with an accuracy of 89.8%, while CDC selectes 10,602 samples with an accuracy of 94.9%. In the 5 classes with the lowest number of correct predictions, FreeMatch selectes 5,833 samples with an accuracy of 38.87%, whereas CDC selectes 6,011 samples with an accuracy of 44.52%.

Fig. 10 shows the category-specific view of selection preferences on CIFAR-20, where FreeMatch's lower thresholds in high-confidence areas introduce more wrong pseudo-labels, impacting performance improvements.

## C.2  STANDARD DEVIATION FOR DEEP CLUSTERING METHODS

Table 10: The clustering performance ACC, NMI, ARI (mean±std %) and calibration error ECE (mean±std %) of various deep clustering methods trained on CIFAR-10, CIFAR-20 and STL-10. All experiments are conducted over 8 different runs.

| Method | CIFAR-10 | | | | CIFAR-20 | | | | STL-10 | | | |
|---|---|---|---|---|---|---|---|---|---|---|---|---|
| | ACC↑ | NMI↑ | ARI↑ | ECE↓ | ACC↑ | NMI↑ | ARI↑ | ECE↓ | ACC↑ | NMI↑ | ARI↑ | ECE↓ |
| MoCo-v2 | 78.9±2.3 | 75.3±2.5 | 61.4±4.0 | - | 47.8±2.0 | 53.9±1.8 | 23.2±1.9 | - | 70.3±4.0 | 69.4±1.8 | 45.5±2.4 | - |
| Simsiam | 72.3±7.7 | 71.5±3.2 | 57.1±5.3 | - | 35.3±1.3 | 38.1±3.0 | 19.4±1.2 | - | 55.0±3.2 | 61.9±3.1 | 36.6±2.0 | - |
| BYOL | 65.1±4.7 | 70.1±1.6 | 54.8±3.1 | - | 33.3±1.6 | 41.6±4.6 | 19.0±1.4 | - | 59.1±5.5 | 67.7±3.0 | 43.4±4.4 | - |
| CC | 82.5±3.6 | 75.2±2.0 | 70.3±3.2 | 8.4±3.4 | 46.4±2.3 | 48.8±1.1 | 31.1±1.5 | 25.8±2.2 | 82.9±7.7 | 76.0±5.2 | 71.8±7.9 | 10.3±7.4 |
| SCAN-2 | 87.3±1.6 | 81.3±1.4 | 77.5±2.2 | 7.4±1.8 | 49.7±1.7 | 53.0±1.1 | 35.2±1.8 | 36.6±1.7 | 88.6±0.9 | 79.6±0.7 | 77.6±1.2 | 5.2±1.1 |
| SCAN-3 | 91.0±1.2 | 83.6±1.5 | 82.1±2.2 | 5.9±1.1 | 51.1±1.9 | 52.7±1.3 | 36.4±1.9 | 39.3±1.8 | 91.8±0.3 | 84.1±0.4 | 83.4±0.6 | 5.7±0.5 |
| CDC-Cal | **94.3±0.6** | **88.5±0.7** | **88.2±1.1** | **1.3±0.2** | **59.4±1.3** | **60.2±0.6** | **44.9±1.1** | **3.5±1.2** | **93.1±0.1** | **86.0±0.2** | **85.6±0.3** | **1.1±0.2** |

Table 11: The clustering performance ACC, NMI, ARI (mean±std %) and calibration error ECE (mean±std %) of various deep clustering methods trained on ImageNet-10, ImageNet-Dogs and Tiny-ImageNet. All experiments are conducted over 8 different runs.

| Method | ImageNet-10 | | | | ImageNet-Dogs | | | | Tiny-ImageNet | | | |
|---|---|---|---|---|---|---|---|---|---|---|---|---|
| | ACC↑ | NMI↑ | ARI↑ | ECE↓ | ACC↑ | NMI↑ | ARI↑ | ECE↓ | ACC↑ | NMI↑ | ARI↑ | ECE↓ |
| MoCo-v2 | 67.4±8.6 | 63.1±6.5 | 45.7±11.1 | - | 65.4±1.9 | 65.7±2.6 | 51.0±2.8 | - | 25.1±1.2 | 42.6±1.3 | 10.6±1.2 | - |
| Simsiam | 78.8±3.7 | 81.8±5.8 | 70.5±8.2 | - | 49.9±5.0 | 51.8±6.9 | 33.6±6.7 | - | 17.4±3.1 | 34.8±2.6 | 7.5±1.5 | - |
| BYOL | 71.8±5.8 | 77.1±4.9 | 60.2±8.0 | - | 59.8±8.1 | 63.7±9.0 | 42.9±7.9 | - | 12.8±3.1 | 30.7±2.8 | 5.3±1.5 | - |
| CC | 91.9±2.6 | 88.8±2.2 | 86.3±3.9 | 6.7±2.5 | 63.9±5.2 | 60.3±3.7 | 49.5±5.3 | 20.6±4.2 | 13.6±0.7 | 41.0±4.3 | 5.7±0.2 | 4.6±0.7 |
| SCAN-2 | 93.3±3.0 | 88.5±2.0 | 87.8±3.9 | 5.1±3.0 | 65.2±4.0 | 62.2±2.4 | 51.8±3.4 | 25.8±3.7 | 27.1±0.4 | 49.9±3.4 | 14.2±0.6 | 23.3±2.1 |
| SCAN-3 | 94.8±3.1 | 91.3±2.2 | 90.7±4.2 | 3.7±2.9 | 72.3±4.0 | 68.9±2.8 | 60.6±4.0 | 20.2±4.1 | 24.9±0.9 | 39.8±0.6 | 12.5±0.6 | 49.2±1.4 |
| CDC-Cal | **97.2±0.1** | **92.9±0.3** | **93.8±0.2** | **0.9±0.1** | **75.5±2.0** | **73.7±1.5** | **66.1±2.2** | **10.4±1.6** | **33.7±0.5** | **47.6±0.2** | **19.9±0.2** | **11.1±0.5** |

From the Tabs. 10-11, we can observe that our method has much smaller variance than most compared methods, while at the same time, produces the best clustering performance and calibration performance. This is because the compared methods usually employ the random initialization on the clustering head, making model training unstable. While our method proposes a novel initialization strategy, utilizing feature prototypes from the pretrained model to initialize the clustering head and the calibration head. This approach significantly reduces the variance in our method's performance as evidenced in Fig.4.

## C.3  TRANSFER OF HYPERPARAMETERS ACROSS DATASETS

We apply the hyperparameters from ImageNet10 (detailed in Sec. 4.1) to the 50, 100, and 200 subsets of ImageNet and compared them with the SCAN method. We conduct the fair comparison under the same backbone network (ResNet-50) and the same subset divisions. For SCAN, we test the trained model from the SCAN code repository [2].

**Results**. The results in Tab. 12 demonstrate a robust improvement in calibration error metrics with our method, showing an average reduction of 8.4%. Meanwhile, the ACC of our method consistently surpasses that of the SCAN method.

Table 12: The clustering performance ACC, NMI, ARI (%) and calibration error ECE (%) of SCAN and CDC on 50, 100, and 200 subsets of ImageNet.

| Method | ImageNet50 | | ImageNet100 | | ImageNet200 | |
|---|---|---|---|---|---|---|
| | ACC↑ | ECE↓ | ACC↑ | ECE↓ | ACC↑ | ECE↓ |
| SCAN-2 | 75.1 | 15.7 | 66.2 | 21.9 | 56.3 | 28.1 |
| SCAN-3 | 76.8 | 14.2 | 68.9 | 18.8 | 58.1 | 25.8 |
| CDC-Cal | **77.8** | **7.5** | **71.2** | **13.0** | **61.2** | **13.2** |

---

[2]https://github.com/wvangansbeke/Unsupervised-Classification

## C.4 GENERAL APPLICABILITY OF CDC AS A SELF-LABELING STAGE

We apply CDC as a post-clustering self-labeling stage to models like SeCu and SCAN. The results presented in Tab. 13 that incorporating CDC following SCAN and SeCu not only enhances clustering accuracy but also significantly decreases the Expected Calibration Error (ECE). This proves the general applicability of our method.

Table 13: Changes in clustering performance metrics (ACC, NMI, ARI %) and Expected Calibration Error (ECE %) after 100 epochs of training across the CIFAR-10 and CIFAR-20 datasets.

| Method | CIFAR-10 | | | | CIFAR-20 | | | |
|---|---|---|---|---|---|---|---|---|
| | ACC↑ | NMI↑ | ARI↑ | ECE↓ | ACC↑ | NMI↑ | ARI↑ | ECE↓ |
| SCAN-2 +CDC | +1.4 | +2.3 | +2.2 | -3.2 | +2.6 | +4.5 | +3.9 | -6.8 |
| SCAN-3 +CDC | +1.2 | +2.6 | +2.4 | -4.3 | +2.0 | +4.2 | +3.0 | -11.9 |
| SeCu-Size +CDC | +1.9 | +2.6 | +3.1 | -0.5 | +3.4 | +3.1 | +3.4 | -10.8 |
| SeCu +CDC | +0.3 | +0.2 | +0.5 | -3.7 | +1.1 | +0.9 | +1.2 | -6.8 |

## C.5 OOD DETECTION FOR DEEP CLUSTERING METHODS

**Datasets**: We employ CIFAR-10 and CIFAR-20 as in-distribution (ID) datasets when mixing data from unrelated datasets. We use Near-OOD datasets (CIFAR-20 and Tiny ImageNet for CIFAR-10; CIFAR-10 and Tiny ImageNet for CIFAR-20) and Far-OOD datasets (SVHN, Textures, Places365).

**Procedure**: For the methods of CC, SCAN, SPICE, SeCu, and CDC (Ours), we use max softmax scores (MSP) of ID and OOD samples to assess the model's performance on AUROC and FPR95 metrics.

Table 14: The AUROC (%) and FPR95 (%) of various deep clustering methods trained on CIFAR-10 for out-of-distribution (OOD) detection.

| OOD→ | CIFAR-100 | | Tiny-ImageNet | | SVHN | | Textures | | Places365 | |
|---|---|---|---|---|---|---|---|---|---|---|
| Method | AUROC↑ | FPR95↓ | AUROC↑ | FPR95↓ | AUROC↑ | FPR95↓ | AUROC↑ | FPR95↓ | AUROC↑ | FPR95↓ |
| CC | 89.9 | 55.3 | 88.0 | 55.2 | 94.4 | 34.9 | 91.3 | 43.2 | 93.2 | 39.9 |
| SCAN-2 | 87.3 | 66.2 | 89.1 | 59.6 | 93.6 | 41.5 | 91.4 | 50.3 | 91.8 | 52.5 |
| SCAN-3 | 88.6 | 64.2 | 92.2 | 49.8 | 95.2 | 33.2 | 92.8 | 47.0 | 93.2 | 46.7 |
| SPICE-2 | 57.0 | 80.8 | 57.1 | 80.6 | 60.5 | 73.9 | 58.1 | 78.7 | 59.4 | 76.1 |
| SPICE-3 | 85.4 | 70.9 | 90.4 | 54.2 | 93.8 | 38.7 | 90.8 | 52.5 | 91.4 | 52.9 |
| SeCu-Size | 77.1 | 80.1 | 80.6 | 77.7 | 81.8 | 74.5 | 80.7 | 84.5 | 79.5 | 79.6 |
| SeCu | 86.8 | 65.8 | 91.5 | 52.0 | 92.1 | 51.3 | 91.9 | 53.3 | 91.4 | 55.8 |
| CDC-Cal | **93.0** | **37.8** | **94.8** | **27.4** | **97.8** | **12.4** | **95.9** | **23.5** | **96.1** | **23.0** |

Table 15: The AUROC (%) and FPR95 (%) of various deep clustering methods trained on CIFAR-20 for out-of-distribution (OOD) detection.

| OOD→ | CIFAR-10 | | Tiny-ImageNet | | SVHN | | Textures | | Places365 | |
|---|---|---|---|---|---|---|---|---|---|---|
| Method | AUROC↑ | FPR95↓ | AUROC↑ | FPR95↓ | AUROC↑ | FPR95↓ | AUROC↑ | FPR95↓ | AUROC↑ | FPR95↓ |
| CC | 69.9 | 86.2 | 77.2 | 77.7 | 85.8 | 63.4 | 76.7 | 77.0 | 80.7 | 72.6 |
| SCAN-2 | 69.3 | 88.5 | 79.8 | 73.8 | 82.3 | 70.9 | 83.2 | 70.0 | 80.5 | 75.2 |
| SCAN-3 | 68.2 | 89.1 | 82.4 | 71.4 | 86.1 | 66.5 | 82.8 | 70.7 | 82.4 | 74.4 |
| SPICE-2 | 51.4 | 92.9 | 52.7 | 90.2 | 54.2 | 87.3 | 53.4 | 88.8 | 53.2 | 89.2 |
| SPICE-3 | 70.2 | 86.4 | 85.0 | 74.6 | 87.8 | 68.1 | 85.2 | 73.8 | 83.7 | 76.6 |
| SeCu-Size | 63.8 | 92.1 | 67.8 | 92.7 | 68.0 | 87.9 | 71.6 | 90.4 | 68.1 | 89.7 |
| SeCu | 73.5 | 83.9 | 82.3 | 74.9 | 86.8 | 74.2 | 82.9 | 77.0 | 82.8 | 76.1 |
| CDC-Cal | **76.8** | 82.6 | **85.7** | **64.6** | **92.2** | **43.8** | **87.8** | **60.6** | **87.4** | **64.0** |

**Results**:As shown in Tab. 14 and Tab. 15, our method demonstrates superior OOD performance compared to comparative methods, indicating the significant role of calibration in rejecting unknown samples. For instance, CDC trained on CIFAR-10 achieved a 2.9% increase in AUROC and a 19.5% decrease in FPR95 compared to the most effective baseline method.

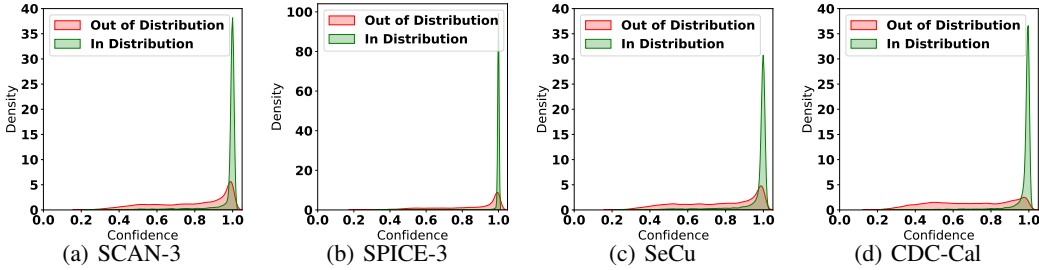

(a) SCAN-3     (b) SPICE-3     (c) SeCu     (d) CDC-Cal

Figure 11: The OOD detection for different deep clustering methods trained on CIFAR-10. The out-of-distribution(OOD) dataset is CIFAR-100. CDC-Cal demonstrates a relatively stronger separation between in-distribution and OOD samples.

We also plot the density-confidence in Fig. 11 to illustrate that the OOD regions of our method are more concentrated in areas of low confidence, demonstrating enhanced separation ability between in-distribution and OOD samples.

