# OpenReview forum: "Towards Calibrated Deep Clustering Network"
_ICLR.cc/2025/Conference — ICLR 2025 Poster_

### Official Review · Reviewer_1qud · 2024-10-29

**Soundness:** 3
**Presentation:** 1
**Contribution:** 2
**Rating:** 5
**Confidence:** 3

**Summary:**

This paper proposes calibrated deep clustering, a method to calibrate the confidence of deep clustering networks. Specifically, it proposes a separate calibration head alongside the traditional clustering head to optimize the learning and prediction confidence of clusters. This approach leads to better-calibrated models as it ensures that the calibration head consistently achieves lower errors compared to the clustering head. The authors provide a theoretical proof to demonstrate that the calibration error from the calibration head is less than or equal to that of the clustering head, supporting the validity of their method.

**Strengths:**

1. The authors provide a comprehensive comparison of their method against 14 prior works. They have also tested the method on 6 datasets, which is a sufficient number to validate their approach. The experimental results clearly demonstrate that CDC achieves significantly better performance than previous methods. The experimental design, including both the main experiments and ablation studies, is rigorous and well-justified.
2. The figures are well-designed and effectively highlight the advantages of the proposed approach. Both the presentation of the results and the visuals are compelling and easy to interpret.
3. The authors offer sound theoretical proof showing that the proposed method achieves a lower expected calibration error and penalizes only unreliable regions, adding to the method’s credibility.
4. The motivation behind the work is well-justified. Both the introduction and abstract are clearly written and provide a strong foundation for the paper.

**Weaknesses:**

1. The authors should reconsider the structure of the paper. In my view, the methods section should first focus on the network and training design, followed by secondary aspects like initialization strategies. While initialization techniques can improve performance and speed up convergence, they are supplementary rather than a primary contribution. Therefore, dedicating excessive space to this, even with theoretical justifications, is unnecessary. Additionally, the experimental results section is too brief. I recommend condensing the methods description and moving a more detailed analysis of results from the appendix into the main body of the paper.

2. The notation is inconsistent and confusing. Several terms are introduced without adequate mathematical explanation. For example, $p_i^{clu}$ is first mentioned in Line 230 as "the prediction of $x_i$ by the clustering head" but lacks a formal mathematical definition. I suggest including an equation, such as defining the clustering head as $g(\theta_{clu},\cdot)$ and expressing $p_i^{clu} = g(\theta_{clu}, x_i)$. A similar issue with $p_i^{cal}$ in Line 235. When introducing the proposed pipeline, the authors should clearly guide the reader through the entire process, from input sample $x$ to the final loss function, in a step-by-step manner in the methods section.

3. The title of Section 3 should be 'Proposed Method'.

Overall, the paper presents promising results and good analysis. However, the presentation of the methodology is lacking, which makes the paper difficult to follow. Revising the organization and clarity of the exposition is neccessary.

**Questions:**

1. The 'classifier $g(\theta, \cdot)$' is mentioned only in the Notation and Algorithm 1. However its role in the proposed pipeline remains unclear to me. Can you clarify its specific function within the pipeline? Is this term meant to generally refer to both the clustering head and calibration head?
2. Can you provide a detailed algorithm outlining how prediction is done?
3. In Equation 4, is the total loss simply the sum of the two individual losses? Is there a hyperparameter that controls the weighting between them? Are the two losses on similar scales? Additionally, is the use of the negative entropy loss inspired by prior works? It would be helpful to see an ablation study that removes this term and compare the results.
4. How does the algorithm perform in the presence of out-of-distribution (OOD) samples? Specifically, could you provide a Density-confidence plot as in figure 3, when OOD samples are introduced in the dataset similar? One possible way to generate OOD samples is by adding random Gaussian noise to selected samples or by mixing data from irrelevant datasets.

---

> ### Author Response · Authors · 2024-11-22
>
> Thank you for your valuable comments.
>
> ---
>
> **Weakness 1: Improvement for Paper Structure and Organization**
>
> Thank you for your constructive suggestions regarding the structure of the paper. We have reorganized the paper in the revised version. Specifically, in Section 3, we first give an overview on the dual-head framework , and  introduce the calibration head and clustering head in sequence. Then we introduce the initialization strategy, the overall training scheme and prediction method.
>
> Furthermore, we have moved an ablation study of "Mini-cluster Number K" from the appendix B.5 of the original paper into the main body of the revised paper to provide a comprehensive evaluation.
>
> We also need to point out that the initialization strategy is important or our work. As evidenced in Figure 4 of the original paper, the initialization technique we proposed promotes the overall performance and stability of the dual-head network.
>
> ---
>
> **Weakness 2: Inconsistent and Ambiguous Notation**
>
> The flowchart in Figure 2 of the original paper can provide a more intuitive explanation. We will **add formal expressions** in the revised version:
>
> - We will modify the notation section (Line 141-147) as: $f (\varTheta; x_i)$ be the feature extractor, $g(θ_{clu};\cdot)$ and $g(θ_{cal};\cdot)$ be the clustering head and the calibration head, $\sigma(\cdot) $ be the softmax function.
> - New definitions of strong augmentation function $\mathcal {A} (\cdot) $and weak augmentation function $\mathcal {W} (\cdot)$.
> - In Line 230 and Line 235, the clustering head and the calibration head respectively predict _un-augmented_ samples as: $p_i^{clu} = \sigma(g(\theta_{clu}; f(\varTheta; x_i)) ) $ and $ p_i^{cal} =  \sigma(g(\theta_{cal}; f(\varTheta; x_i))) $.
> - In Line 303, the calibration head provides predictions for _weakly_ augmented samples: $p_i^{w cal} = \sigma(g(\theta_{cal}; f(\varTheta; \mathcal{W}(x_i)))) $.
> - In Line 323, the clustering head provides predictions for _strongly_ augmented samples: $ p_i^{s clu} =  \sigma(g(\theta_{clu}; f(\varTheta; \mathcal{A}(x_i)))) $.
>
> These changes will make the methodology clearer and more accessible to readers.
>
> ---
>
> **Weakness 3: Title of Section 3**
>
> We will change the title of Section 3 to "Proposed Method" as suggested.
>
> ---
>
> **Question 1: Clarification of 'classifier g(θ,⋅)'**
>
> The classifier $g(θ;\cdot)$ indeed refers to both the clustering head $g(θ_{clu};\cdot)$ and the calibration head $g(θ_{cal};\cdot)$. Its role is to project the feature embeddings $f (\varTheta; x_i)$ into the respective output spaces, specifically, $g(θ_{clu};f (\varTheta; x_i))$ and $g(θ_{cal};f (\varTheta; x_i))$ for the clustering head and calibration head respectively.
>
> We agree simply using $g(θ;\cdot)$ is unclear. We have replaced $g(θ;\cdot)$ with $g(θ_{clu};\cdot)$ and $g(θ_{cal};\cdot)$ in the revised paper, including in the Notation section and Algorithm 1.
>
> The function **pipeline** of our network is: $x_i$ (input) $\rightarrow$ $f (\varTheta; x_i)$ (backbone)  $\rightarrow$  $g(θ_{clu};f (\varTheta; x_i))$ (clustering head) and $g(θ_{cal};f (\varTheta; x_i))$  (calibration head).
>
> ---
>
> **Question 2: Prediction Algorithm**
>
> As we mentioned in “Final Prediction” (subsection 3.4) in Line 333-334 of the original paper, the final prediction of i-th sample $x_i$ is given by the calibration head. We will add the formula to clarify the presentation, i.e.,  $p_i^{cal} =  \sigma(g(θ_{cal}; f(\varTheta; x_i)))$.

---

> ### Author Response · Authors · 2024-11-22
>
> **Question 3: Loss in Equation 4**
>
> Equation 4 is the sum of the two individual losses, $L_{cal}$ and  $L_{en}$. There is no additional hyperparameter for weighting the two losses because they are designed to operate on similar scales. Specifically, $ L_{cal}$ denotes the **average** entropy per sample, while $ L_{en} $represents the negative entropy of the **average** prediction across all samples.
>
> The use of the negative entropy loss $ L_{en} $ is inspired by prior works on deep clustering methods (e.g., CC, SCAN, SPICE and SeCu). It helps **avoid assigning all samples to a single cluster**, promoting a more balanced clustering structure.
>
> Following your suggestions, we conduct an ablation study that varies the scale of the coefficient $w_{en}$ for $ L_{en} $, i.e., $L=L_{cal}+w_{en} L_{en}$. The results presented in Table Q3 below. The findings indicate that the proposed model is robust to $w_{en}$, so we fixed it at $w_{en} = 1$ for simplicity in the experiments. Moreover, removing the negative entropy loss $L_{en}$ will degrade the clustering performance, especially on CIFAR-20.
>
> **Table Q3: Influence of coefficient $w_{en}$ on clustering accuracy (ACC %) and expected calibration error (ECE %) across STL-10 and CIFAR-20 datasets.**
>
> | Dataset  | $w_{en}$ | 0.0  | 0.1  | 0.5  | 1.0      | 5.0  | 10.0     |
> | -------- | -------- | ---- | ---- | ---- | -------- | ---- | -------- |
> | STL-10   | ACC↑     | 92.9 | 93.0 | 92.8 | 93.0     | 93.1 | **93.2** |
> |          | ECE↓     | 1.1  | 1.0  | 1.3  | **0.9**  | 0.9  | 1.0      |
> | CIFAR-20 | ACC↑     | 57.0 | 57.7 | 58.8 | **61.7** | 58.2 | 56.9     |
> |          | ECE↓     | 2.4  | 2.7  | 1.8  | 4.9      | 2.7  | **1.7**  |
>
> ---
>
> **Question 4: Out-of-Distribution (OOD) Behavior**
>
> Thank you for your question, which is crucial for evaluating our method's ability to withstand distribution shifts. Following your suggestion, we have conducted additional experiments. The specific settings and results are as follows:
>
> **Settings**
>
> - **Datasets**: We employ CIFAR-10 and CIFAR-20 as in-distribution (ID) datasets  when mixing data from unrelated datasets. We use **Near-OOD datasets** (CIFAR-20 and Tiny ImageNet for CIFAR-10; CIFAR-10 and Tiny ImageNet for CIFAR-20) and **Far-OOD datasets** (SVHN, Textures, Places365).
> - **Procedure**: For the methods of CC, SCAN, SPICE, SeCu, and CDC (Ours), we use **max softmax scores** (MSP) of ID and OOD samples to assess the model's performance on **AUROC** and **FPR95** metrics.
>
> **Results**:
> As shown in Table Q4-1 and Table Q4-2 below, our method demonstrates superior OOD performance compared to comparative methods, indicating the significant role of calibration in rejecting unknown samples. For instance, CDC trained on CIFAR-10 achieved a 2.9% increase in AUROC and a 19.5% decrease in FPR95 compared to the most effective baseline method.
>
> We also plot the density-confidence in Figure 11 of the revised paper to illustrate that the OOD regions of our method are more concentrated in areas of low confidence, demonstrating enhanced separation ability between in-distribution and OOD samples.

---

> ### Author Response · Authors · 2024-11-22
>
> **Table Q4-1: The AUROC (%) and FPR95 (%) of various deep clustering methods trained on CIFAR-10 for out-of-distribution (OOD) detection.**
>
> | OOD→      |    CIFAR-100    |                   |  Tiny-ImageNet  |                   |      SVHN       |                   |    Textures     |                   |    Places365    |                   |
> | --------- | :-------------: | :---------------: | :-------------: | :---------------: | :-------------: | :---------------: | :-------------: | :---------------: | :-------------: | :---------------: |
> | CIFAR-10  | AUROC$\uparrow$ | FPR95$\downarrow$ | AUROC$\uparrow$ | FPR95$\downarrow$ | AUROC$\uparrow$ | FPR95$\downarrow$ | AUROC$\uparrow$ | FPR95$\downarrow$ | AUROC$\uparrow$ | FPR95$\downarrow$ |
> | CC        |      89.9       |       55.3        |      88.0       |       55.2        |      94.4       |       34.9        |      91.3       |       43.2        |      93.2       |       39.9        |
> | SCAN2     |      87.3       |       66.2        |      89.1       |       59.6        |      93.6       |       41.5        |      91.4       |       50.3        |      91.8       |       52.5        |
> | SCAN3     |      88.6       |       64.2        |      92.2       |       49.8        |      95.2       |       33.2        |      92.8       |       47.0        |      93.2       |       46.7        |
> | SPICE2    |      57.0       |       80.8        |      57.1       |       80.6        |      60.5       |       73.9        |      58.1       |       78.7        |      59.4       |       76.1        |
> | SPICE3    |      85.4       |       70.9        |      90.4       |       54.2        |      93.8       |       38.7        |      90.8       |       52.5        |      91.4       |       52.9        |
> | SeCu-Size |      77.1       |       80.1        |      80.6       |       77.7        |      81.8       |       74.5        |      80.7       |       84.5        |      79.5       |       79.6        |
> | SeCu      |      86.8       |       65.8        |      91.5       |       52.0        |      92.1       |       51.3        |      91.9       |       53.3        |      91.4       |       55.8        |
> | CDC       |    **93.0**     |     **37.8**      |    **94.8**     |     **27.4**      |    **97.8**     |     **12.4**      |    **95.9**     |     **23.5**      |    **96.1**     |     **23.0**      |
>
> **Table Q4-2: The AUROC (%) and FPR95 (%) of various deep clustering methods trained on CIFAR-20 for out-of-distribution (OOD) detection.**
>
> | OOD→      |    CIFAR-10     |                   |  Tiny-ImageNet  |                   |      SVHN       |                   |    Textures     |                   |    Places365    |                   |
> | --------- | :-------------: | :---------------: | :-------------: | :---------------: | :-------------: | :---------------: | :-------------: | :---------------: | :-------------: | :---------------: |
> | CIFAR-20  | AUROC$\uparrow$ | FPR95$\downarrow$ | AUROC$\uparrow$ | FPR95$\downarrow$ | AUROC$\uparrow$ | FPR95$\downarrow$ | AUROC$\uparrow$ | FPR95$\downarrow$ | AUROC$\uparrow$ | FPR95$\downarrow$ |
> | CC        |      69.9       |       86.2        |      77.2       |       77.7        |      85.8       |       63.4        |      76.7       |       77.0        |      80.7       |       72.6        |
> | SCAN2     |      69.3       |       88.5        |      79.8       |       73.8        |      82.3       |       70.9        |      83.2       |       70.0        |      80.5       |       75.2        |
> | SCAN3     |      68.2       |       89.1        |      82.4       |       71.4        |      86.1       |       66.5        |      82.8       |       70.7        |      82.4       |       74.4        |
> | SPICE2    |      51.4       |       92.9        |      52.7       |       90.2        |      54.2       |       87.3        |      53.4       |       88.8        |      53.2       |       89.2        |
> | SPICE3    |      70.2       |       86.4        |      85.0       |       74.6        |      87.8       |       68.1        |      85.2       |       73.8        |      83.7       |       76.6        |
> | SeCu-Size |      63.8       |       92.1        |      67.8       |       92.7        |      68.0       |       87.9        |      71.6       |       90.4        |      68.1       |       89.7        |
> | SeCu      |      73.5       |       83.9        |      82.3       |       74.9        |      86.8       |       74.2        |      82.9       |       77.0        |      82.8       |       76.1        |
> | CDC       |    **76.8**     |     **82.6**      |    **85.7**     |     **64.6**      |    **92.2**     |     **43.8**      |    **87.8**     |     **60.6**      |    **87.4**     |     **64.0**      |

---

> > ### Comment · Reviewer_1qud · 2024-11-25
> >
> > Thanks for the detailed rebuttal. I am satisfied with the revised manuscript and appreciate the detailed experiment results. I have thus raised my score.

---

> ### Author Response · Authors · 2024-11-29
>
> Dear Reviewer **1qud**,
>
> Thank you once again for your time and efforts in reviewing our paper. We appreciate your valuable feedback. Other reviewers have indicated that their concerns have been addressed and have acknowledged our contributions. Although you mentioned that "*I am satisfied with the revised manuscript and appreciate the detailed experiment results*", we noticed that your score is still **not positive**. Thus, we would like to know if there are any additional concerns we can address. We would be more than happy to provide further responses.
>
> We appreciate your consideration.
>
> Best regards,
>
> The authors

---

> > ### Author Response · Authors · 2024-12-02
> > **We are wating for your feedback. Thanks!**
> >
> > Dear Reviewer **1qud**,
> >
> > As the Reviewer-Author discussion phase is drawing to a close, we kindly ask you to review our revisions and responses once more and reconsider your rating. We eagerly anticipate your feedback. Thank you.
> >
> > Best regards,
> >
> > The Authors

---

### Official Review · Reviewer_KQtX · 2024-11-02

**Soundness:** 2
**Presentation:** 3
**Contribution:** 1
**Rating:** 5
**Confidence:** 3

**Summary:**

This paper proposes a dual head deep clustering model that can effectively calibrate the estimated confidence and the actual accuracy.

**Strengths:**

This paper proposes a dual head deep clustering model that can effectively calibrate the estimated confidence and the actual accuracy.

**Weaknesses:**

According to the discussion between the authors and other reviewers, I altered my rating.

**Questions:**

No

---

> ### Author Response · Authors · 2024-11-22
>
> While no specific questions were raised, we remain open to any suggestions or clarifications that would help improve the presentation and impact of the work. Thank you for your constructive feedback and for taking the time to review our paper.

---

> ### Author Response · Authors · 2024-11-27
>
> Dear Reviewer **KQtX**,
>
> Thank you for dedicating your time and effort to reviewing our manuscript. Initially, we noted that you **did not** provide any comments in the Weaknesses and Questions section. However, we observed that you **quietly** adjusted the rating from 6 to 5 without indicating any specific issues, which left us feeling perplexed.
>
> We have noticed that you have now included a comment in the Weaknesses section stating, "According to the discussion between the authors and other reviewers, I altered my rating." We kindly request that you please elaborate on your specific concerns. As in the current discussion period, all the other reviewers say their concerns have been addressed, and some have raised their scores accordingly. For example, both Reviewer yP9a and Review bpNw raised their score from 6 to 8.
>
> It is through constructive feedback and clear communication that each of us can contribute effectively as professional reviewers, thus enhancing the quality of our community.
>
> Thank you.

---

### Official Review · Reviewer_8jfQ · 2024-11-02

**Soundness:** 3
**Presentation:** 3
**Contribution:** 3
**Rating:** 6
**Confidence:** 4

**Summary:**

The paper considers calibration in the context of deep clustering, more specifically pseudo-label self-training based deep clustering. The authors propose an architecture consisting of a backbone (trained in a self-supervised manner) and two heads, where one is the common clustering head, while the other is a calibration head. The calibration head obtains calibrated confidences by overclustering the data, computing the average prediction for each of the clusters, and then aligning the predictions of the calibration head for members of the clusters to the clusters average prediction. The better calibrated predictions can then be leveraged for the pseudo-label selection. Beyond the calibration head, the authors propose a new initialization strategy for clustering (and calibration) heads that does ensure initialization that resembles the k-means performance on the backbone features as well as a mechanism to dynamically select the threshold for pseudo-label selection for each class. The proposed approach is evaluated on six datasets, obtaining promising performance and lower expected calibration error compared to prior deep clustering baselines.

**Strengths:**

- The paper is overall well written and the contributions presented in a clear and well-motivated manner.

- Considering calibration in deep clustering is growing more important as state-of-the-art methods rely increasingly on pseudo-label self-training and has been not been addressed previously to the reviewers knowledge.

- The proposed approaches are intuitive and effective as indicated by the empirical results and the ablation study.

- Empirical results are supported by theoretic insights and analysis.

**Weaknesses:**

Hyperparameter selection is challenging in a clustering setting when no labeled data is available. It is therefore beneficial to have approaches that are robust with respect to hyperparameters. Currently, the sensitivity to the added hyperparameters is not discussed and relative specific hyperparameters are defined for different datasets (B, K) or not provided (H). How were these hyperparameters selected and how sensitive is the model to these choices. Do the authors have insights on how these should be chosen in the typical clustering setting where labeled data is unavailable?

One weakness of the work is that results are currently reported over single runs and without information about variability. Deep clustering approaches tend to be less stable than their supervised counterparts and some more information on the robustness of these results would therefore be beneficial.

The claim in the abstract regarding surpassing state-of-the-art deep clustering methods by 10x in term of expected calibration error should be revised to clarify that this is compared to only some of the baselines and on some of the considered datasets.

**Questions:**

Could the authors comment on the sensitivity with respect to the hyperparameters as well as on the variability in results over multiple runs?

Should the subscripts be switched in Theorem 1? So $E_F[Conf^{cal}]$ <= $E_F[Conf^{clu}]$ and  $E_T[Conf^{cal}]$ = $E_T[Conf^{clu}]$?

---

> ### Author Response · Authors · 2024-11-22
>
> Thank you for your valuable comments.
>
> ---
>
> **Weakness 1 & Question 1: Hyperparameter Selection and Sensitivity**
> The influences of K are shown in Appendix B.5 of the original paper. On the whole, we selected hyperparameters B, K, and H based on empirical evaluation. Specifically:
>
> **Batch size and Number of mini-clusters (B, K)**: As demonstrated in Table W1-1 below, our model exhibits considerable robustness to variations within a reasonable range for B and K, especially on challenging datasets such as CIFAR-20. Variations of K by ±20% and B by ±50% only result in the changes in overall accuracy of ±1.4%.
>
> **Table W1-1: Influence of (B, K) on clustering accuracy (ACC %) and expected calibration error (ECE %) across STL-10 and CIFAR-20 datasets.**
>
> |          | B    | 500  | 500  | 500  | 1000 | 1000 | 1000 | 1500 | 1500 | 1500 |      |
> | -------- | ---- | ---- | ---- | ---- | ---- | ---- | ---- | ---- | ---- | ---- | ---- |
> |          | K    | -20% | 0%   | +20% | -20% | 0%   | +20% | -20% | 0%   | +20% | Std  |
> | STL-10   | ACC  | 93.0 | 93.1 | 93.2 | 93.1 | 93.0 | 93.1 | 92.8 | 92.8 | 93.1 | 0.1  |
> |          | ECE  | 1.1  | 1.2  | 1.6  | 1.2  | 0.9  | 1.2  | 1.2  | 1.0  | 0.8  | 0.2  |
> | CIFAR-20 | ACC  | 58.1 | 61.0 | 60.2 | 58.8 | 61.7 | 62.6 | 59.1 | 60.0 | 59.2 | 1.4  |
> |          | ECE  | 6.9  | 7.6  | 3.6  | 4.9  | 4.9  | 4.6  | 4.4  | 1.4  | 3.9  | 1.7  |
>
> - **Insight for B**: We constrain GPU memory usage to within 9GB. For datasets with smaller image size, larger batch sizes are utilized (e.g., B=1000 for CIFAR-10 and CIFAR-20, which have image sizes of 32x32). Conversely, smaller batch sizes are employed for datasets with larger image sizes (e.g., B=500 for ImageNet-10 and ImageNet-Dogs, which have image sizes of 224x224).
> - **Insight for K**: K is tuned empirically according to the complexity of the dataset. For challenging datasets with over-confident predictions, a lower K value is needed to increase the confidence penalty, with an empirical ratio of $B/K \le5$. Conversely, for simpler datasets, a higher K value is necessary to reduce the confidence penalty, supported by an empirical ratio of $B/K \ge10$. More analysis on the selection of K can be found in Sec. 4.3 of the revised paper.
>
> **Hidden units (H)**: The choice of 512-dim is commonly used in MLP, and provides a good balance between initialization complexity and performance, as Table W1-2 shows empirically.
>
> **Table W1-2: Influence of MLP Hidden layer size on clustering accuracy (ACC %) and expected calibration error (ECE %) across STL-10 and CIFAR-20 datasets.**
>
> | Dataset  | #Hidden         | 256  | 384  | 512  | 640  | 768  |
> | -------- | --------------- | ---- | ---- | ---- | ---- | ---- |
> | STL-10   | ACC$\uparrow$   | 93.2 | 93.1 | 93.0 | 92.6 | 92.2 |
> |          | ECE$\downarrow$ | 1.7  | 1.1  | 0.9  | 0.6  | 1.0  |
> | CIFAR-20 | ACC$\uparrow$   | 61.7 | 60.6 | 61.7 | 58.3 | 57.1 |
> |          | ECE$\downarrow$ | 3.7  | 4.1  | 4.9  | 3.5  | 2.5  |
>
> ---
>
> **Weakness 2 & Question 1: Variability in Results Across Multiple Runs**
>
> The experimental results in Table 2 of the original paper were obtained from a single run under a random seed of 1024. We have depicted the training process, including variance, for both our method and comparative methods in Figure 4 of the original paper. To further analyze the performance of the models across all datasets, we conducted **three runs** with different random seeds, and the results are presented in the Table W2-1 and Table W2-2 below.
>
> From the Table W2-1 and Table W2-2, we can observe that our method has much smaller variance than most compared methods, while at the same time, produces the best clustering performance and calibration performance. This is because the compared methods usually employ the random initialization on the clustering head, making model training unstable. While our method proposes a novel initialization strategy, utilizing feature prototypes from the pretrained model to initialize the clustering head and the calibration head. This approach significantly reduces the variance in our method's performance as evidenced in Figure 4 of the original paper.
>
> The results and analyses has been added in Appendix C.2 of the revised paper.
>
> ---
>
> **Weakness 3: Clarification of Abstract Claim Regarding 10x Improvement in ECE**
> Thanks for your suggestions, we will revise this statement to "by 5x on average" to avoid exaggeration.
>
> ---
>
> **Question 2: Subscripts in Theorem 1**
> Yes, the subscripts should be switched as follows:
>
> - $ \mathbb{E}\_F[Conf\_{cal}] \leq \mathbb{E}\_F[Conf\_{clu}] $
> - $ \mathbb{E}\_T[Conf\_{cal}] = \mathbb{E}\_T[Conf\_{clu}] $

---

> ### Author Response · Authors · 2024-11-22
>
> **Table W2-1: The clustering performance ACC, NMI, ARI (mean±std %) and calibration error ECE (mean±std %) of various deep clustering methods trained on CIFAR-10, CIFAR-20 and STL-10.**
>
> |  Method   |     CIFAR-10     |                  |                  |                 |     CIFAR-20     |                  |                  |                 |      STL-10      |                  |                  |                 |
> | :-------: | :--------------: | :--------------: | :--------------: | :-------------: | :--------------: | :--------------: | :--------------: | :-------------: | :--------------: | :--------------: | :--------------: | :-------------: |
> |           |  ACC$\uparrow$   |  NMI$\uparrow$   |  ARI$\uparrow$   | ECE$\downarrow$ |  ACC$\uparrow$   |  NMI$\uparrow$   |  ARI$\uparrow$   | ECE$\downarrow$ |  ACC$\uparrow$   |  NMI$\uparrow$   |  ARI$\uparrow$   | ECE$\downarrow$ |
> |    CC     |     80.4±3.4     |     74.2±2.0     |     68.5±3.1     |    10.4±3.0     |     45.7±2.6     |     48.8±1.3     |     30.8±1.6     |    26.7±2.4     |     79.9±8.6     |     74.1±5.9     |     69.1±9.1     |    13.2±8.3     |
> |  SCAN-2   |     86.7±1.9     |     81.4±1.8     |     77.4±2.8     |     8.0±2.1     |     49.9±2.2     |     53.0±1.2     |     35.2±2.1     |    36.5±2.3     |     88.2±1.0     |     79.4±0.7     |     77.1±1.3     |     5.6±1.3     |
> |  SCAN-3   |     91.3±1.4     |     83.9±1.9     |     82.7±2.7     |     5.6±1.3     |     50.8±2.1     |     52.4±1.2     |     36.2±1.8     |    39.6±2.0     |     91.7±0.3     |     84.0±0.4     |     83.1±0.6     |     5.9±0.5     |
> | SeCu-Size |     88.6±0.9     |     80.3±0.9     |     78.5±1.8     |     4.4±0.8     |     50.7±1.7     |     51.8±1.9     |     35.9±1.6     |    15.3±1.7     |     80.2±0.8     |     69.4±0.9     |     63.7±1.3     |     9.8±0.8     |
> |   SeCu    |     91.8±0.4     |     85.7±0.4     |     84.3±0.5     |     5.8±0.4     |     52.6±0.2     |     56.5±0.3     |     39.6±0.2     |    41.3±1.2     |     83.0±0.6     |     72.7±0.6     |     68.4±0.9     |     7.1±0.6     |
> |  CDC-Cal  | **94.3**±**0.5** | **88.5**±**0.5** | **88.2**±**0.9** | **1.2**±**0.2** | **59.3**±**1.6** | **59.9**±**0.7** | **44.8**±**1.3** | **3.1**±**1.3** | **93.1**±**0.1** | **86.0**±**0.1** | **85.7**±**0.1** | **1.0**±**0.1** |
>
> **Table W2-2: The clustering performance ACC, NMI, ARI (mean±std %) and calibration error ECE (mean±std %) of various deep clustering methods trained on ImageNet-10, ImageNet-Dogs and Tiny-ImageNet.**
>
> | Method  |   ImageNet-10    |                  |                  |                 |  ImageNet-Dogs   |                  |                  |                  |  Tiny-ImageNet   |                  |                  |                 |
> | :-----: | :--------------: | :--------------: | :--------------: | :-------------: | :--------------: | :--------------: | :--------------: | :--------------: | :--------------: | :--------------: | :--------------: | :-------------: |
> |         |  ACC$\uparrow$   |  NMI$\uparrow$   |  ARI$\uparrow$   | ECE$\downarrow$ |  ACC$\uparrow$   |  NMI$\uparrow$   |  ARI$\uparrow$   | ECE$\downarrow$  |  ACC$\uparrow$   |  NMI$\uparrow$   |  ARI$\uparrow$   | ECE$\downarrow$ |
> |   CC    |     92.5±3.3     |     89.1±2.8     |     87.0±4.9     |     6.1±3.1     |     62.2±5.5     |     59.4±4.3     |     47.7±6.0     |     22.9±2.9     |     13.3±0.9     |     39.7±5.0     |     5.6±0.1      |   **4.4**±0.8   |
> | SCAN-2  |     93.6±2.8     |     88.6±1.8     |     88.1±3.6     |     4.6±2.9     |     67.3±2.9     |     63.1±2.2     |     53.3±2.7     |     23.6±2.0     |     27.1±0.4     |     48.7±4.0     |     14.3±0.7     |    24.1±2.4     |
> | SCAN-3  |     95.3±2.8     |     92.0±1.8     |     91.6±3.8     |     3.3±2.8     |     74.8±2.6     |     69.9±2.5     |     62.3±3.6     |     17.0±2.5     |     24.7±1.0     |     39.9±1.0     |     12.7±0.7     |    50.1±1.0     |
> | CDC-Cal | **97.2**±**0.1** | **93.0**±**0.2** | **93.9**±**0.1** | **0.8**±**0.0** | **75.8**±**2.4** | **73.9**±**1.8** | **66.3**±**2.6** | **10.3**±**1.9** | **33.8**±**0.3** | **47.6**±**0.3** | **19.9**±**0.2** |  11.1±**0.4**   |

---

> > ### Comment · Reviewer_8jfQ · 2024-11-25
> >
> > Thank you for the detailed response to my questions and providing additional clarifications.
> >
> > I appreciate the insights that the authors have provided regarding the hyperparameters. However, some concerns remain, as prior works such as SCAN and SPICE employ fixed hyperparameters across datasets to avoid empirical tuning (using labeled data) in an unsupervised setting. That said, since the main contribution is on the calibration, where gaps are sufficiently large to not be affected, I will maintain my rating.

---

> > > ### Author Response · Authors · 2024-11-26
> > >
> > > We appreciate the reviewer bringing up the comparison to prior works such as SCAN and SPICE, which employ fixed hyperparameters across datasets. While our approach involves some hyperparameter choices, we want to clarify that:
> > >
> > > **1. Fixed threshold is not as effective as expected**. While fixed hyperparameters are appealing for simplicity, this approach may fail to address dataset-specific challenges, particularly when overconfidence issues vary in magnitude. For the SCAN method on Tiny-ImageNet, if we use the same high threshold as other datasets, such as 0.99, the ACC significantly decreases (from 27.6% to 12.2%). Although our reproduction adjusts it to 0.80, there is still a decline (from 27.6% to 25.8%).
> > >
> > > **2. Our primary parameter is the number of mini-clusters (K), and our tuning aimed at achieving optimal calibration error rather than clustering performance.**
> > >
> > > - If the calibration module is removed, there is **free of manual setting of hyper-parameters** for our method (K is removed and other hyper-parameters are fixed), and the performance of the CDC remains competitive. These results are demonstrated in our ablation study III in Section 4.2, where CDC utilizes overconfident network predictions to adaptively adjust thresholds without relying on any hyperparameters. The corresponding results are redefined as CDC-w/o Cal in the following table. The experimental results demonstrate that, even without calibration hyperparameters, the model's clustering performance still surpasses that of the comparison methods.
> > >
> > > **Table C-2: Performance comparison of different methods. CDC-Cal denotes CDC with the calibration module retained. CDC-w/o Cal denotes CDC with the calibration module removed.**
> > >
> > >
> > >
> > > |               | CIFAR-10 |          |          |         | CIFAR-20 |          |          |         |  STL-10  |          |          |         |
> > > | ------------- | :------: | :------: | :------: | :-----: | :------: | :------: | :------: | :-----: | :------: | :------: | :------: | :-----: |
> > > | Method        |   ACC    |   NMI    |   ARI    |   ECE   |   ACC    |   NMI    |   ARI    |   ECE   |   ACC    |   NMI    |   ARI    |   ECE   |
> > > | CC            |   85.2   |   76.9   |   72.8   |   6.2   |   42.4   |   47.1   |   28.4   |  29.7   |   80.0   |   72.7   |   67.7   |  11.9   |
> > > | SCAN-2        |   84.1   |   79.3   |   74.1   |  10.9   |   50.0   |   52.2   |   34.7   |  37.1   |   87.0   |   78.8   |   75.6   |   7.4   |
> > > | SCAN-3        |   90.3   |   82.5   |   80.8   |   6.7   |   51.2   |   51.2   |   35.6   |  39.0   |   91.4   |   83.6   |   82.5   |   6.6   |
> > > | SPICE-2       |   84.4   |   73.5   |   70.9   |  15.4   |   47.6   |   44.3   |   30.3   |  52.3   |   89.6   |   80.4   |   79.2   |  10.1   |
> > > | SPICE-3       |   91.5   |   85.4   |   83.4   |   7.8   |   58.4   |   57.6   |   42.2   |  40.6   |   93.0   |   86.1   |   85.5   |   6.3   |
> > > | SeCu-Size     |   90.0   |   83.2   |   81.5   |   8.1   |   52.9   |   54.9   |   38.4   |  13.1   |   80.2   |   69.4   |   63.1   |   9.9   |
> > > | SeCu          |   92.6   |   86.2   |   85.4   |   4.9   |   52.7   |   56.7   |   39.7   |  41.8   |   83.6   |   73.3   |   69.3   |   6.5   |
> > > | CDC-w/o   Cal |   93.9   |   88.0   |   87.5   |   2.3   |   59.7   |   61.3   |   45.3   |  31.6   |   92.6   |   85.3   |   84.9   |   5.1   |
> > > | CDC-Cal       | **94.9** | **89.3** | **89.5** | **1.1** | **61.7** | **60.9** | **46.6** | **4.9** | **93.0** | **85.8** | **85.6** | **0.9** |
> > >
> > > - The fixed calibration hyperparameters can also lead to **excellent clustering and calibration performance empirically with theoretically guaranteed** (shown in Theorem 2 of the paper).
> > >
> > >   - **Settings**. We applied the hyperparameters from ImageNet10 (detailed in "Implementation Details" in Section 4 of the revised paper) to the 50, 100, and 200 subsets of ImageNet and compared them with the SCAN method. We conducted the fair comparison under the same backbone network (ResNet-50) and the same subset divisions. For SCAN, we downloaded and tested the trained model from the SCAN code repository.
> > >   - **Results**. The results below demonstrate a robust improvement in calibration error metrics with our method, showing an average reduction of 8.4%. Meanwhile, the ACC of our method consistently surpasses that of the SCAN method.
> > >
> > >   **Table C-3: Performance of SCAN and CDC on 50, 100, and 200 subsets of ImageNet.**
> > >
> > >   |         | ImageNet50 |         | ImageNet100 |          | ImageNet200 |          |
> > >   | ------- | ---------- | ------- | ----------- | -------- | ----------- | -------- |
> > >   |         | ACC        | ECE     | ACC         | ECE      | ACC         | ECE      |
> > >   | SCAN-2  | 75.1       | 15.7    | 66.2        | 21.9     | 56.3        | 28.1     |
> > >   | SCAN-3  | 76.8       | 14.2    | 68.9        | 18.8     | 58.1        | 25.8     |
> > >   | CDC-Cal | **77.8**   | **7.5** | **71.2**    | **13.0** | **61.2**    | **13.2** |

---

### Official Review · Reviewer_yP9a · 2024-11-02

**Soundness:** 3
**Presentation:** 3
**Contribution:** 3
**Rating:** 8
**Confidence:** 4

**Summary:**

The authors identify the problem of overconfident predictions in deep clustering methods. To overcome this problem, they propose a dual head design with a dedicated calibration head aside the usual clustering head to improve deep clustering performance and calibration. Further, they propose an initialization strategy for the dual heads that improves stability when transitioning between the pretraining and deep clustering phases.

**Strengths:**

**Originality**
- The authors identify and investigate the issue of overconfidence in deep clustering methods. Based on their findings they propose a novel Calibrated Deep Clustering (CDC) method.  This is a quite an interesting angle to improve deep clustering methods and connects the field to confidence calibration methods from (semi)-supervised learning.

**Quality**
- The authors provide extensive experiments across different benchmark data sets and deep clustering methods. Further, they made the extra effort of reimplementing SeCu, SCAN and CC to obtain a unified benchmarking environment increasing the fairness of the comparison. I would highly encourage the authors to release their code with the full benchmarking environment, as this will benefit the deep clustering community. Additionally, the authors provide ablation studies and a hyperparameter analysis for their algorithm CDC.

**Clarity**
- The paper is well written, and all design decisions are clearly motivated

**Significance**
- Their method CDC combines SCAN’s self-labeling procedure with a calibration head.  As self-labeling is widely used and their method does not make specific assumptions about the underlying deep clustering method, I believe this method can impact many deep clustering methods going forward.

**Weaknesses:**

- Unclear reporting of experimental results.  The authors fail to mention if they report average performance results in Table 2 and other figures, or if they report the results of a single (best) run. As many deep clustering methods have high variability across different random seeds it would be important to report standard deviations as well, e.g., SCAN reports standard deviations of up to 3% for CIFAR-20.  Currently, it is not possible to judge whether the proposed method CDC is stable across multiple runs, because no standard deviations are reported and it is not mentioned if several runs are conducted at all.

- The hyperparameter for mini-clusters K needs to be specified quite differently for each data set ranging from 40 to 1000. Further, K does not seem to be related to the ground truth number of clusters of each data set and the authors do not provide any unsupervised heuristic that can be used to set K. The same holds true for the batch size B.

- The authors do not specify how they tuned their MoCo-v2 backbone. Further, no hyperparameters for the pretraining are reported. This is an important aspect that needs to be reported for reproducibility.

**Questions:**

- What exactly is reported in Table 2? Are these the best or average results over multiple runs? If yes, how many runs were conducted and what are the standard deviations across runs? See also the corresponding weakness described above.

- Hyperparameter sensitivity analysis w.r.t. K in Figure 8 shows that CDC is stable for different K for CIFAR-20 and STL-10, but it is set quite different for each data set.  Is there a heuristic to set K or does K need tuning w.r.t. the ground truth labels for each data set? See also the corresponding weakness described above.

- Is the MoCo-v2 pretraining used for CC, SCAN and SeCu as well or just for your method? Based on your description it is not entirely clear whether you just use the same ResNet-34 backbone for these methods or if you use MoCo-v2 pretraining for them as well.

- Please clarify how you conducted the MoCo-v2 pretraining and which hyperparameters you used for each data set.

- Based on my understanding your method could be used as a dedicated self-labeling stage on top of any deep clustering method, similar to how SCAN's self-labeling is currently used to improve clustering performance after the joint deep clustering stage. I would be interested to see how your method would improve, e.g., SeCu after its deep clustering stage. This would further illustrate the general applicability of your method and increase its significance for the deep clustering community.

---

> ### Author Response · Authors · 2024-11-22
>
> Thank you for your valuable comments.
>
> ---
>
> **Weakness 1 & Question 1: Reporting of Experimental Results**
>
> The experimental results in Table 2 of the original paper were obtained from a single run under a random seed of 1024. We have depicted the training process, including variance, for both our method and comparative methods in Figure 4 of the original paper. To further analyze the performance of the models across all datasets, we conducted **three runs** with different random seeds, and the results are presented in the Table W1-1 and Table W1-2 below.
>
> From the Table W1-1 and Table W1-2, we can observe that our method has much smaller variance than most compared methods, while at the same time, produces the best clustering performance and calibration performance. This is because the compared methods usually employ the random initialization on the clustering head, making model training unstable. While our method proposes a novel initialization strategy, utilizing feature prototypes from the pretrained model to initialize the clustering head and the calibration head. This approach significantly reduces the variance in our method's performance as evidenced in Figure 4 of the original paper.
>
> The results and analyses has been added in Appendix C.2 of the revised paper.

---

> ### Author Response · Authors · 2024-11-22
>
> **Table W1-1: The clustering performance ACC, NMI, ARI (mean±std %) and calibration error ECE (mean±std %) of various deep clustering methods trained on CIFAR-10, CIFAR-20 and STL-10.**
>
> |  Method   |     CIFAR-10     |                  |                  |                 |     CIFAR-20     |                  |                  |                 |      STL-10      |                  |                  |                 |
> | :-------: | :--------------: | :--------------: | :--------------: | :-------------: | :--------------: | :--------------: | :--------------: | :-------------: | :--------------: | :--------------: | :--------------: | :-------------: |
> |           |  ACC$\uparrow$   |  NMI$\uparrow$   |  ARI$\uparrow$   | ECE$\downarrow$ |  ACC$\uparrow$   |  NMI$\uparrow$   |  ARI$\uparrow$   | ECE$\downarrow$ |  ACC$\uparrow$   |  NMI$\uparrow$   |  ARI$\uparrow$   | ECE$\downarrow$ |
> |    CC     |     80.4±3.4     |     74.2±2.0     |     68.5±3.1     |    10.4±3.0     |     45.7±2.6     |     48.8±1.3     |     30.8±1.6     |    26.7±2.4     |     79.9±8.6     |     74.1±5.9     |     69.1±9.1     |    13.2±8.3     |
> |  SCAN-2   |     86.7±1.9     |     81.4±1.8     |     77.4±2.8     |     8.0±2.1     |     49.9±2.2     |     53.0±1.2     |     35.2±2.1     |    36.5±2.3     |     88.2±1.0     |     79.4±0.7     |     77.1±1.3     |     5.6±1.3     |
> |  SCAN-3   |     91.3±1.4     |     83.9±1.9     |     82.7±2.7     |     5.6±1.3     |     50.8±2.1     |     52.4±1.2     |     36.2±1.8     |    39.6±2.0     |     91.7±0.3     |     84.0±0.4     |     83.1±0.6     |     5.9±0.5     |
> | SeCu-Size |     88.6±0.9     |     80.3±0.9     |     78.5±1.8     |     4.4±0.8     |     50.7±1.7     |     51.8±1.9     |     35.9±1.6     |    15.3±1.7     |     80.2±0.8     |     69.4±0.9     |     63.7±1.3     |     9.8±0.8     |
> |   SeCu    |     91.8±0.4     |     85.7±0.4     |     84.3±0.5     |     5.8±0.4     |     52.6±0.2     |     56.5±0.3     |     39.6±0.2     |    41.3±1.2     |     83.0±0.6     |     72.7±0.6     |     68.4±0.9     |     7.1±0.6     |
> |  CDC-Cal  | **94.3**±**0.5** | **88.5**±**0.5** | **88.2**±**0.9** | **1.2**±**0.2** | **59.3**±**1.6** | **59.9**±**0.7** | **44.8**±**1.3** | **3.1**±**1.3** | **93.1**±**0.1** | **86.0**±**0.1** | **85.7**±**0.1** | **1.0**±**0.1** |
>
> **Table W1-2: The clustering performance ACC, NMI, ARI (mean±std %) and calibration error ECE (mean±std %) of various deep clustering methods trained on ImageNet-10, ImageNet-Dogs and Tiny-ImageNet.**
>
> | Method  |   ImageNet-10    |                  |                  |                 |  ImageNet-Dogs   |                  |                  |                  |  Tiny-ImageNet   |                  |                  |                  |
> | :-----: | :--------------: | :--------------: | :--------------: | :-------------: | :--------------: | :--------------: | :--------------: | :--------------: | :--------------: | :--------------: | :--------------: | :--------------: |
> |         |  ACC$\uparrow$   |  NMI$\uparrow$   |  ARI$\uparrow$   | ECE$\downarrow$ |  ACC$\uparrow$   |  NMI$\uparrow$   |  ARI$\uparrow$   | ECE$\downarrow$  |  ACC$\uparrow$   |  NMI$\uparrow$   |  ARI$\uparrow$   | ECE$\downarrow$  |
> |   CC    |     92.5±3.3     |     89.1±2.8     |     87.0±4.9     |     6.1±3.1     |     62.2±5.5     |     59.4±4.3     |     47.7±6.0     |     22.9±2.9     |     13.3±0.9     |     39.7±5.0     |     5.6±0.1      |     4.4±0.8      |
> | SCAN-2  |     93.6±2.8     |     88.6±1.8     |     88.1±3.6     |     4.6±2.9     |     67.3±2.9     |     63.1±2.2     |     53.3±2.7     |     23.6±2.0     |     27.1±0.4     |     48.7±4.0     |     14.3±0.7     |     24.1±2.4     |
> | SCAN-3  |     95.3±2.8     |     92.0±1.8     |     91.6±3.8     |     3.3±2.8     |     74.8±2.6     |     69.9±2.5     |     62.3±3.6     |     17.0±2.5     |     24.7±1.0     |     39.9±1.0     |     12.7±0.7     |     50.1±1.0     |
> | CDC-Cal | **97.2**±**0.1** | **93.0**±**0.2** | **93.9**±**0.1** | **0.8**±**0.0** | **75.8**±**2.4** | **73.9**±**1.8** | **66.3**±**2.6** | **10.3**±**1.9** | **33.8**±**0.3** | **47.6**±**0.3** | **19.9**±**0.2** | **11.1**±**0.4** |

---

> ### Author Response · Authors · 2024-11-22
>
> **Weakness 2 & Question 2: Hyperparameter Sensitivity and Selection of \( K \)**
>
> **Insight for K**: The influences of min-clusters K on the performance have been shown in the Appendix B.5 of the original paper. We have the following insight to tune the value of K. For challenging datasets with over-confident predictions, a lower K value is needed to increase the confidence penalty, with an empirical ratio of $B/K \le5$. Conversely, for simpler datasets, a higher K value is necessary to reduce the confidence penalty, supported by an empirical ratio of $B/K \ge10$.
>
> See more analysis in Appendix B.5 of the original paper.
>
> **Batch size and Number of mini-clusters (B, K)**: As demonstrated in Table W2 below, our model exhibits considerable robustness to variations within a reasonable range for B and K, especially on challenging datasets such as CIFAR-20. Variations of K by ±20% and B by ±50% only result in the changes in overall accuracy of ±1.4%.
>
> **Table W2: Influence of (B, K) on clustering accuracy (ACC %) and expected calibration error (ECE %) across STL-10 and CIFAR-20 datasets.**
>
> |          | B    | 500  | 500  | 500  | 1000 | 1000 | 1000 | 1500 | 1500 | 1500 |      |
> | -------- | ---- | ---- | ---- | ---- | ---- | ---- | ---- | ---- | ---- | ---- | ---- |
> |          | K    | -20% | 0%   | +20% | -20% | 0%   | +20% | -20% | 0%   | +20% | Std  |
> | STL-10   | ACC  | 93.0 | 93.1 | 93.2 | 93.1 | 93.0 | 93.1 | 92.8 | 92.8 | 93.1 | 0.1  |
> |          | ECE  | 1.1  | 1.2  | 1.6  | 1.2  | 0.9  | 1.2  | 1.2  | 1.0  | 0.8  | 0.2  |
> | CIFAR-20 | ACC  | 58.1 | 61.0 | 60.2 | 58.8 | 61.7 | 62.6 | 59.1 | 60.0 | 59.2 | 1.4  |
> |          | ECE  | 6.9  | 7.6  | 3.6  | 4.9  | 4.9  | 4.6  | 4.4  | 1.4  | 3.9  | 1.7  |
>
> ---
>
> **Weakness 3 & Question 3 & Question 4: MoCo-v2 Pretraining Details**
>
> Details on how to train the backbone are provided in Appendix "B.1 IMPLEMENTATION DETAILS" (Lines 854-878) of the original paper. **Moreover, the code has been submitted to the Supplementary Material, where all the configurations are specified**.
>
> **MoCo-v2 Hyperparameters** (Lines 854-865): The Appendix B.1 details settings applicable to all datasets used in this study. We also provide **parameter script files** for all training stages across six datasets in the code files, facilitating reproducibility. Furthermore, we plan to open-source both the MoCo-v2 pre-trained models and the trained models.
>
> **Usage of MoCo-v2** (Line 871): We clarify that MoCo-v2 pre-training is utilized in the initial phase of the CC, SCAN, and CDC (ours) methods but not in SeCu due to its distinct training approach.
>
> **Usage of Backbone** (Lines 858, 866-869, 871-876): Unless otherwise specified, ResNet-34 is employed as the backbone across all stages of the CC, SCAN, SeCu, and CDC (ours), as well as in the comparative methods.
>
> In the revised paper, we have listed the parameter settings in Appendix B.1 to enhance the readability and reproducibility of the paper.
>
> ---
>
> **Question 5: General Applicability of CDC as a Self-Labeling Stage**
> Following your suggestions, we applied CDC as a post-clustering self-labeling stage to models like SeCu and SCAN. The results presented in Table Q5 demonstrate that incorporating CDC following SCAN and SeCu not only enhances clustering accuracy but also significantly decreases the Expected Calibration Error (ECE). This proves the general applicability of our method. Thanks again for your suggestions.
>
> **Table Q5: Changes in clustering performance metrics (ACC, NMI, ARI %) and Expected Calibration Error (ECE %) after 100 epochs of training across the CIFAR-10 and CIFAR-20 datasets.**
>
> | CIFAR-10   | ACC↑ | NMI↑ | ARI↑ | ECE↓ | CIFAR-20   | ACC↑ | NMI↑ | ARI↑ | ECE↓  |
> | ---------- | ---- | ---- | ---- | ---- | ---------- | ---- | ---- | ---- | ----- |
> | SCAN-2+CDC | +1.4 | +2.3 | +2.2 | -3.2 | SCAN-2+CDC | +2.6 | +4.5 | +3.9 | -6.8  |
> | SCAN-3+CDC | +1.2 | +2.6 | +2.4 | -4.3 | SCAN-3+CDC | +2.0 | +4.2 | +3.0 | -11.9 |
> | SeCu+CDC   | +0.3 | +0.2 | +0.5 | -3.7 | SeCu+CDC   | +1.1 | +0.9 | +1.2 | -6.8  |

---

> > ### Comment · Reviewer_yP9a · 2024-11-25
> >
> > Thank you for the additional experiments and clarifications. I appreciate the extra effort in applying CDC to other deep clustering methods, the results look promising as well, highlighting the significance of the presented work. However, I will keep my initial score and decided against giving a higher score, due to two reasons. First, while the variation between the three runs for CDC is low, it would still be more convincing to use 5-10 runs with different seeds as is usually done in the literature, e.g., SeCu used eight different runs. Second, I agree with reviewer `8jfQ` that the hyperparameter tuning for each dataset is a concern in practice, even though it is commonly done in research where we have access to labels. Unfortunately, there is no agreed upon standard in the literature and different published methods did their evaluation differently. One experiment that might alleviate this concern is to check whether the hyperparameters transfer to related data sets. For example, you can show that good hyperparameter settings for CDC on different subsets of ImageNet are very similar to each other.
> >
> > All that said, this is still solid work and the paper should be accepted, but the proposed experiments could be included in a camera ready version.

---

> > > ### Author Response · Authors · 2024-11-26
> > >
> > > We greatly appreciate the reviewer’s thoughtful feedback and recognition of the contributions of our work. Below, we address the two concerns raised:
> > >
> > > **1. Runs with Multiple Seeds**:
> > > We agree that increasing the number of runs with different random seeds would provide additional confidence in the robustness of our method.
> > >
> > > Following your suggestion, we conducted eight different runs on CIFAR-20 and STL-10, and cited the results from SCAN (10 runs), SeCu-Size (8 runs), and SeCu-Entropy (8 runs) from their original papers.
> > >
> > > The results in Table C-1 show that the variance of our method are significantly smaller than the comparison methods (1.3% on CIFAR-20 compared to SCAN's 2.7%, and 0.1% on STL-10 compared to SCAN's 1.9%). Our method not only achieves the lowest variance but also exhibits the best clustering performance, benefiting from a prototype-based initialization strategy that enhances model robustness and training stability.
> > >
> > > In Section C.2 of the revised paper, we have added the above experiments  and analyses. For the final version, we will extend all experiments to include results averaged over 8 runs with different seeds.
> > >
> > > **Table C-1: Results from multiple runs on CIFAR-20 and STL-10. We cited SCAN (10 runs), SeCu-Size (8 runs), and SeCu-Entropy (8 runs) from their original papers.**
> > >
> > > |                     |   CIFAR-20    |               |               |                 |    STL-10     |               |               |                 |
> > > | ------------------- | :-----------: | :-----------: | :-----------: | :-------------: | :-----------: | :-----------: | :-----------: | :-------------: |
> > > | Method              | ACC$\uparrow$ | NMI$\uparrow$ | ARI$\uparrow$ | ECE$\downarrow$ | ACC$\uparrow$ | NMI$\uparrow$ | ARI$\uparrow$ | ECE$\downarrow$ |
> > > | SCAN-2              |   42.2±3.0    |   44.1±1.0    |   26.7±1.3    |        -        |   75.5±2.0    |   65.4±1.2    |   59.0±1.6    |        -        |
> > > | SCAN-3              |   45.9±2.7    |   46.8±1.3    |   30.1±2.1    |        -        |   76.7±1.9    |   68.0±1.2    |   61.6±1.8    |        -        |
> > > | SeCu-Size (mean)    |      50       |     50.7      |      35       |        -        |     80.2      |     69.4      |     63.9      |        -        |
> > > | SeCu-Size (best)    |     51.6      |     51.6      |      36       |        -        |     81.4      |     70.7      |     65.7      |        -        |
> > > | SeCu-Entropy (mean) |     49.9      |     50.6      |     34.1      |        -        |     79.5      |     68.7      |      63       |        -        |
> > > | SeCu-Entropy (best) |     51.2      |     51.4      |     34.9      |        -        |     80.5      |     69.9      |     64.4      |        -        |
> > > | CDC-Cal             | **59.4±1.3**  | **60.2±0.6**  | **44.9±1.1**  |   **3.5±1.2**   | **93.1±0.1**  |  **86±0.2**   | **85.6±0.3**  |   **1.1±0.2**   |

---

> > > ### Author Response · Authors · 2024-11-26
> > >
> > > **2. Hyperparameter Transferability**:
> > >
> > > We appreciate the suggestion to demonstrate hyperparameter transferability, as this would further address practical concerns regarding dataset-specific tuning.
> > >
> > > Our primary parameter is the number of mini-clusters (K), and our tuning aimed at achieving optimal calibration error rather than clustering performance.
> > >
> > > - If the calibration module is removed, there is **free of manual setting of hyper-parameters** for our method (K is removed and other hyper-parameters are fixed), and the performance of the CDC remains competitive. These results are demonstrated in our ablation study III in Section 4.2, where CDC utilizes overconfident network predictions to adaptively adjust thresholds without relying on any hyperparameters. The corresponding results are redefined as CDC-w/o Cal in the following table. The experimental results demonstrate that, even without calibration hyperparameters, the model's clustering performance still surpasses that of the comparison methods.
> > >
> > > **Table C-2: Performance comparison of different methods. CDC-Cal denotes CDC with the calibration module retained. CDC-w/o Cal denotes CDC with the calibration module removed.**
> > >
> > > |               | CIFAR-10 |          |          |         | CIFAR-20 |          |          |         |  STL-10  |          |          |         |
> > > | ------------- | :------: | :------: | :------: | :-----: | :------: | :------: | :------: | :-----: | :------: | :------: | :------: | :-----: |
> > > | Method        |   ACC    |   NMI    |   ARI    |   ECE   |   ACC    |   NMI    |   ARI    |   ECE   |   ACC    |   NMI    |   ARI    |   ECE   |
> > > | CC            |   85.2   |   76.9   |   72.8   |   6.2   |   42.4   |   47.1   |   28.4   |  29.7   |   80.0   |   72.7   |   67.7   |  11.9   |
> > > | SCAN-2        |   84.1   |   79.3   |   74.1   |  10.9   |   50.0   |   52.2   |   34.7   |  37.1   |   87.0   |   78.8   |   75.6   |   7.4   |
> > > | SCAN-3        |   90.3   |   82.5   |   80.8   |   6.7   |   51.2   |   51.2   |   35.6   |  39.0   |   91.4   |   83.6   |   82.5   |   6.6   |
> > > | SPICE-2       |   84.4   |   73.5   |   70.9   |  15.4   |   47.6   |   44.3   |   30.3   |  52.3   |   89.6   |   80.4   |   79.2   |  10.1   |
> > > | SPICE-3       |   91.5   |   85.4   |   83.4   |   7.8   |   58.4   |   57.6   |   42.2   |  40.6   |   93.0   |   86.1   |   85.5   |   6.3   |
> > > | SeCu-Size     |   90.0   |   83.2   |   81.5   |   8.1   |   52.9   |   54.9   |   38.4   |  13.1   |   80.2   |   69.4   |   63.1   |   9.9   |
> > > | SeCu          |   92.6   |   86.2   |   85.4   |   4.9   |   52.7   |   56.7   |   39.7   |  41.8   |   83.6   |   73.3   |   69.3   |   6.5   |
> > > | CDC-w/o   Cal |   93.9   |   88.0   |   87.5   |   2.3   |   59.7   |   61.3   |   45.3   |  31.6   |   92.6   |   85.3   |   84.9   |   5.1   |
> > > | CDC-Cal       | **94.9** | **89.3** | **89.5** | **1.1** | **61.7** | **60.9** | **46.6** | **4.9** | **93.0** | **85.8** | **85.6** | **0.9** |
> > >
> > > - The fixed calibration hyperparameters can also lead to **excellent clustering and calibration performance empirically with theoretically guaranteed** (shown in Theorem 2 of the paper).
> > >
> > >   - **Settings**. **We applied the hyperparameters from ImageNet10 (detailed in "Implementation Details" in Section 4 of the revised paper) to the 50, 100, and 200 subsets of ImageNet** and compared them with the SCAN method. We conducted the fair comparison under the same backbone network (ResNet-50) and the same subset divisions. For SCAN, we downloaded and tested the trained model from the SCAN code repository.
> > >   - **Results**. The results below demonstrate a robust improvement in calibration error metrics with our method, showing an average reduction of 8.4%. Meanwhile, the ACC of our method consistently surpasses that of the SCAN method.
> > >
> > >   **Table C-3: Performance of SCAN and CDC on 50, 100, and 200 subsets of ImageNet.**
> > >
> > >   |         | ImageNet50 |         | ImageNet100 |          | ImageNet200 |          |
> > >   | ------- | ---------- | ------- | ----------- | -------- | ----------- | -------- |
> > >   |         | ACC        | ECE     | ACC         | ECE      | ACC         | ECE      |
> > >   | SCAN-2  | 75.1       | 15.7    | 66.2        | 21.9     | 56.3        | 28.1     |
> > >   | SCAN-3  | 76.8       | 14.2    | 68.9        | 18.8     | 58.1        | 25.8     |
> > >   | CDC-Cal | **77.8**   | **7.5** | **71.2**    | **13.0** | **61.2**    | **13.2** |

---

> > > > ### Comment · Reviewer_yP9a · 2024-11-26
> > > >
> > > > Thank you again for the additional experiments in response to the raised concerns. The results look very promising and I will increase my score accordingly. However, for the final version I would urge the authors to conduct all experiments over multiple seeds.

---

> > > > > ### Author Response · Authors · 2024-11-27
> > > > >
> > > > > Thank you again for raising the score. In Appendix Section C.2 of the newly revised paper, we further supplement the variance of some compared methods with 8 different seeds on all the datasets. We can see that our method can produce the best clustering performance with the lowest variance, indicating its robustness.

---

### Official Review · Reviewer_bpNw · 2024-11-03

**Soundness:** 2
**Presentation:** 3
**Contribution:** 3
**Rating:** 8
**Confidence:** 4

**Summary:**

This paper addresses the over-confidence problem in deep clustering, where the predicted confidence for a sample significantly exceeds its actual accuracy. The authors propose a novel calibrated deep clustering framework that consists of a dual-head model, including a calibration head and a clustering head. The calibration head corrects the overconfident predictions from the clustering head, aligning estimated confidence with the model's learning status. The clustering head then selects reliable high-confidence samples for pseudo-label self-training. Additionally, the paper introduces a new network initialization strategy that improves training speed and robustness. The proposed model demonstrates significant improvements over state-of-the-art methods, achieving a 10× reduction in expected calibration error and enhanced clustering accuracy, supported by solid theoretical guarantees.

**Strengths:**

- Novel Approach to Calibration: The introduction of a dual-head model that effectively calibrates confidence estimates addresses a critical gap in deep clustering, enhancing the reliability of predictions.

- Improved Self-Training Mechanism: By leveraging high-confidence samples for pseudo-label self-training, the proposed method improves the quality of the training data, which is crucial for achieving better clustering performance.

- Robust Experimental Validation: The extensive experiments conducted demonstrate the effectiveness of the calibrated deep clustering model, showcasing significant improvements in both expected calibration error and clustering accuracy compared to existing methods, which adds credibility to the proposed approach.

**Weaknesses:**

- This paper would benefit from a more comprehensive literature review. Specifically, it overlooks some closely related works, such as [a,b], which address the over-confidence issue for standard supervised learning or unsupervised domain adaptation (i.e., unlabeled target data clustering with the help of labeled source data). I recommend that the authors include a discussion of these works and its connections to their research. This will help to contextualize their contributions within the existing body of knowledge.

[a] Revisiting the Calibration of Modern Neural Networks. NeurIPS, 2021.

[b] Transferable Calibration with Lower Bias and Variance in Domain Adaptation. NeurIPS, 2020.

- The authors mention that the calibration head adjusts the overconfident predictions of the clustering head to generate prediction confidence that aligns with the model's learning status. However, in the context of completely unsupervised learning and clustering, it is not clear how the model's learning status is determined without any labeled data. I recommend that the authors provide a more detailed explanation of the methodology used to assess the model's learning state in the absence of supervision. This clarification is essential for understanding the effectiveness of the calibration head and its role in improving prediction confidence.

- The authors state that the clustering head dynamically selects reliable high-confidence samples estimated by the calibration head for pseudo-label self-training. However, it would be beneficial to explore whether insights from existing semi-supervised learning (SSL) methods, such as those discussed in [c], could be incorporated to further enhance the self-training process.

Additionally, a critical analysis of current self-training methods is warranted to determine whether they contribute to poor calibration of predictions. It would be valuable to quantify the extent of this calibration issue and discuss its implications on model performance. By addressing these points, the authors could strengthen their argument and provide a more comprehensive understanding of the calibration dynamics in their approach.

[c] Towards Discovering the Effectiveness of Moderately Confident Samples for Semi-Supervised Learning. CVPR, 2022.

- The authors note that existing approaches rely on a fixed threshold to select pseudo labels, which does not account for the model's learning status. This can lead to issues during different training stages: a higher threshold in the early stages may result in selecting fewer and class-imbalanced samples, hindering convergence speed, while in later stages, a fixed threshold may introduce noisy pseudo-labels due to increased predicted confidence.

It would be beneficial for the authors to consider incorporating insights from existing SSL works, such as [d] and its subsequent studies, which have proposed dynamic thresholds and class-wise thresholds. These methods could potentially enhance the adaptability of pseudo-label selection throughout the training process, improving both the quality of the training data and the overall performance of the model. A discussion on how these approaches could be integrated or compared with the proposed methodology would strengthen this work.

[d] FlexMatch: Boosting Semi-Supervised Learning with Curriculum Pseudo Labeling. NeurIPS, 2021.

**Questions:**

See Weaknesses.

---

> ### Author Response · Authors · 2024-11-22
>
> Thank you for your valuable comments.
>
> ---
>
> **Weakness 1: Comprehensive Literature Review**
>
> We appreciate the reviewer’s suggestion to expand the literature review and include discussions of related works addressing the over-confidence issue in supervised learning and unsupervised domain adaptation. The suggested references [a] and [b] are indeed relevant, and we will incorporate them into the revised paper.
>
> **[a] Revisiting the Calibration of Modern Neural Networks (NeurIPS, 2021):** This paper examines calibration in supervised learning and provides a comprehensive empirical analyses of modern neural networks, exploring how architectural and training techniques impact calibration. While [a] primarily addresses calibration in supervised learning, our work extends the concept to unsupervised clustering, where calibration is inherently more challenging due to the absence of labeled data.
>
> **[b] Transferable Calibration with Lower Bias and Variance in Domain Adaptation (NeurIPS, 2020):** This study tackles calibration in the domain adaptation setting, where labeled source data is available. [b] leverages labeled source data to achieve calibration, while our method operates in a fully unsupervised setting. We will highlight these distinctions while addressing the overconfidence issue through targeted calibration techniques.
>
> These discussions will contextualize our contributions within existing researches and clarify the novelty of our approach.
>
> ---
>
> **Weakness 2: Determining the Model’s Learning Status in an Unsupervised Context**
>
> In our approach, the calibration head estimates the learning status by analyzing the confidence distribution of predictions given by the clustering head. Specifically:
>
> - **Intra cluster level**. The calibration head are learned based on the average confidence within mini-clusters, providing a better approximation of the model’s learning status than the confidence of single sample.
> - **Inter cluster level**. Some previous methods estimate the learning status using the average of maximum predicted values across the entire dataset, or employing class-wise averages (as seen in SSL approaches like FlexMatch[R1] and FreeMatch[R2]). Those learning status estimation methods are coarse-grained and may be misled by the possible over-confidence predictions. Our calibration method **alleviates overconfidence** and **operates at a finer granularity**, i.e., we divide embedding space into reliable regions and unreliable regions, and we only trust the confidence predictions in reliable regions to estimate learning status, and calibrate the confidence of the samples in unreliable regions. So our approach offers a more accurate estimation of the state.
>
> [R1] FlexMatch: Boosting Semi-Supervised Learning with Curriculum Pseudo Labeling. NeurIPS, 2021.
>
> [R2] FreeMatch: Self-adaptive Thresholding for Semi-supervised Learning. ICLR, 2023.
>
> The learning status estimated by our method is not perfectly accurate, but it is **effective in practice** and **aligns well with unsupervised settings**. Furthermore, ablation study III in Section 4.2 from the original paper demonstrates that relying on over-confident predictions as a measure of learning state leads to performance degradation, indicating the effectiveness of our learning status estimation method.
>
> ---

---

> ### Author Response · Authors · 2024-11-22
>
> **Weakness 3: Enhancing Self-Training with SSL Insights**
>
> We appreciate the suggestion to incorporate insights from semi-supervised learning (SSL) methods such as [c]. The mentioned work [c] explores more moderately confident samples via Gradient Synchronization Filter (GSF) and Prototype Proximity Filter (PPF) to boost learning, which is conceptually aligned with our approach to pseudo-labeling.
>
> We applied this method to expand our selected sample set, and the experimental results are presented in Table W3 below.
>
> **Table W3: Clustering Performance (ACC, NMI, ARI %) on STL-10 and CIFAR-20.**
>
> |         | STL-10   |          |          | CIFAR-20 |          |          |
> | ------- | -------- | -------- | -------- | -------- | -------- | -------- |
> | Method  | ACC      | NMI      | ARI      | ACC      | NMI      | ARI      |
> | CDC     | 93.0     | 85.8     | 85.6     | 61.7     | 60.9     | 46.6     |
> | CDC+GSF | 93.0     | 85.8     | 85.5     | 61.9     | 61.2     | **46.8** |
> | CDC+PPF | **93.0** | **85.8** | **85.6** | **62.1** | **61.3** | 46.6     |
>
> From the above table, we find that GSF and PPF can only bring marginal improvements in performance. The reason is the **highly overlap in sample selection** between the CDC's strategy and those selected by GSF and PPF,  the expansion of selected samples is less than 5%.
>
> Nevertheless, related techniques hold potential to further enhance our model, which we intend to explore thoroughly in the future.
>
> ---
>
>
>
> **Weakness 3: Impact of Self Training Methods on Calibration**
>
> Calibration is employed to measure the gap between confidence and accuracy. Compared to supervised learning, Self-Training exhibits a more pronounced overconfidence—overfitting some inevitable incorrect pseudo labels—thereby increasing calibration errors, as evidenced in Fig.1 (a)-(c) of the original paper.
>
> In deep clustering, self-supervised methods increase confidence, and at the same time introduce substantial supervised information, leading to improvements in accuracy. Changes in the Expected Calibration Error (ECE) are influenced by both confidence and accuracy. Our experiments reveal that Self-Training degrades calibration in two scenarios, exemplified by SCAN-3 (applying self-labeling to SCAN-2):
>
> 1. Accuracy increases slower than confidence, as demonstrated on CIFAR-20 (ACC +1.2%, CONF +3.1%, ECE +1.9%).
> 2. Accuracy decreases (due to fitting too many noisy labels), as shown on Tiny-ImageNet (ACC -1.8%, CONF +19.5%, ECE +21.4%).
>
> However, improvements in ECE are also observed in some cases, as shown on CIFAR-10 (ACC +6.2%, CONF +2.0%, ECE -4.2%), where the overall confidence increase is limited (CONF 95.0%→97.0%).
>
> The key to model calibration is **narrowing the gap between confidence and accuracy**. Our explored techniques include:
>
> - **Regularization loss (targeting confidence only)**. The methods based on regularization loss reduce overall confidence to improve calibration, where effective calibration does not enhance performance (as evidenced in Section 4.1 "Competitive Failure Rejection Ability" of the original paper).
> - **Our approach (targeting both confidence and accuracy)**. Our method controls the increase in confidence while maintaining accurate predictions in reliable regions, achieving simultaneous improvements in performance and calibration ability.
>
> Therefore, calibration methods should consider relationship between confidence and accuracy to achieve improvements in model performance.

---

> ### Author Response · Authors · 2024-11-22
>
> **Weakness 4: Insights from Dynamic Thresholding in SSL**
>
> Thank you for pointing out relevant work such as FlexMatch [d], which proposes dynamic and class-wise thresholds for pseudo-label selection. Additionally, we incorporate subsequent developments like FreeMatch[R2], which adapts the predefined fixed global threshold of FlexMatch to be adaptive based on the learning status. We explore the application of the thresholding strategies from FlexMatch and FreeMatch in our CDC framework by two approaches (**Apply Directly** and **Integration**).
>
> [R2] FreeMatch: Self-adaptive Thresholding for Semi-supervised Learning. ICLR, 2023.
>
> **Table W4-1: Clustering performance ACC, NMI, ARI(%) and calibration error ECE (%) of Semi-Supervised Learning methods applied directly to self-labeling on STL-10 and CIFAR-20. Sel. represents the proportion of selected samples.**
>
> |           |   CIFAR-20    |               |               |                 |          |    STL-10     |               |               |                 |          |
> | :-------: | :-----------: | :-----------: | :-----------: | :-------------: | :------: | :-----------: | :-----------: | :-----------: | :-------------: | :------: |
> |  Method   | ACC$\uparrow$ | NMI$\uparrow$ | ARI$\uparrow$ | ECE$\downarrow$ | Sel. (%) | ACC$\uparrow$ | NMI$\uparrow$ | ARI$\uparrow$ | ECE$\downarrow$ | Sel. (%) |
> | FlexMatch |     54.9      |     55.6      |     39.2      |      37.4       |   80.2   |     92.6      |     85.6      |     84.8      |       5.8       |   90.6   |
> | FreeMatch |     48.8      |     54.9      |     16.8      |      44.0       |   90.8   |     93.0      |   **85.9**   |     85.6      |       5.1       |   88.3   |
> |    CDC    |   **61.7**   |   **60.9**   |   **46.6**   |    **4.9**     |   55.2   |   **93.0**   |     85.8      |   **85.6**   |    **0.9**     |   89.4   |
>
> **Apply Directly:**
>
> - **Settings**. Consistent with CDC, after pre-training with MoCo-v2 and performing prototype-based initialization, we utilize pseudo-label learning, evaluating the learning state based on the model's output.
> - **Analysis for Table W4-1**. CDC outperforms FlexMatch and FreeMatch on both datasets. The reasons are as follows. FlexMatch and FreeMatch rely on the prediction confidence estimated by a well-calibrated model, which is reasonable in semi-supervised learning (SSL) scenarios where some labels are known. In deep clustering, the model **updates thresholds based on over-confident predictions**, which can lead to the selection of more noisy labels, causing **performance degradation**—this effect is more pronounced in challenging datasets like CIFAR-20. Specifically, selecting 80.2% of the overall samples corresponds to an accuracy of 54.9% in FlexMatch, selecting 90.8% of the overall samples also corresponds to an accuracy of 48.8% in FreeMatch, whereas selecting 55.2% of the overall samples corresponds to an accuracy of 61.7% in CDC.

---

> ### Author Response · Authors · 2024-11-22
>
> **Weakness 4: Insights from Dynamic Thresholding in SSL**
>
> **Table W4-2: Clustering performance ACC, NMI, ARI(%) and calibration error ECE (%) of Semi-Supervised Learning methods integrated into CDC on STL-10 and CIFAR-20. Sel. represents the proportion of selected samples.**
>
> |           |   CIFAR-20    |               |               |                 |          |    STL-10     |               |               |                 |          |
> | :-------: | :-----------: | :-----------: | :-----------: | :-------------: | :------: | :-----------: | :-----------: | :-----------: | :-------------: | :------: |
> |  Method   | ACC$\uparrow$ | NMI$\uparrow$ | ARI$\uparrow$ | ECE$\downarrow$ | Sel. (%) | ACC$\uparrow$ | NMI$\uparrow$ | ARI$\uparrow$ | ECE$\downarrow$ | Sel. (%) |
> | FlexMatch |     57.2      |     59.4      |     42.1      |       6.5       |   3.2    |     92.3      |     84.6      |     84.0      |       2.0       |   91.2   |
> | FreeMatch |     57.8      |     59.1      |     43.6      |     **1.4**     |   64.3   |     92.9      |     85.6      |     85.3      |     **0.7**     |   81.9   |
> |    CDC    |   **61.7**    |   **60.9**    |   **46.6**    |       4.9       |   55.2   |   **93.0**    |   **85.8**    |   **85.6**    |       0.9       |   89.4   |
>
> **Integration:**
>
> - **Settings**. Consistent with the previous settings, only the threshold selection strategy of CDC is replaced by FlexMatch and FreeMatch.
>
> - **Analysis for Table W4-2**. CDC outperforms FlexMatch and FreeMatch on both datasets. The reasons are as follows. Under the predictions of a well-calibrated model, **FlexMatch** requires a **dataset-specific global threshold**. When the global threshold for STL-10 and CIFAR-20 is set to 0.95 as recommended by [d], it results in significantly different sample selection scales (91.2% in STL-10 and 3.2% in CIFAR-20), affecting performance improvements. **FreeMatch** is more robust, similar to CDC, with both methods being free from manually setting thresholds. The difference between FreeMatch and CDC on CIFAR-20 lies in the handling of **head classes** (classes with a large number of selected samples):
>
>   - **FreeMatch** uses local threshold adjustments to modify the overall threshold (the average of the maximum probability values for all samples), with local thresholds after maxNorm ranging from 0 to 1. Thus, the thresholds for head classes are less than the **overall threshold**. Accordingly, FreeMatch will introduce more samples for these classes (and more wrong pseudo-labels).
>   - **CDC** directly selects reliable samples based on calibrated probabilities for each class, ensuring that the thresholds for head classes are **not reduced by the overall threshold**.
>
> - **Selected Samples Analysis in Figure 10**.
>
>   Figure 10 of the revised paper shows the category-specific view of selection preferences on CIFAR-20, where **FreeMatch's lower thresholds in high-confidence areas introduce more wrong pseudo-labels**, impacting performance improvements.
>
>   Specifically, Among the top 5 classes with the highest number of correct predictions, FreeMatch selectes 12,671 samples with an accuracy of 89.8%, while CDC selectes 10,602 samples with an accuracy of 94.9%. In the 5 classes with the lowest number of correct predictions, FreeMatch selectes 5,833 samples with an accuracy of 38.87%, whereas CDC selectes 6,011 samples with an accuracy of 44.52%.
>
> Given that the semi-supervised learning also encounters unlabeled samples, the applicability of semi-supervised methods to deep clustering warrants further investigation. Thanks again for your suggestion.

---

> ### Comment · Reviewer_bpNw · 2024-11-26
> **Incorporate the suggested works for discussions or experiments**
>
> I have reviewed the authors' rebuttal as well as the comments from the other reviewers. I appreciate the thorough discussions, analyses, and experiments presented, which effectively address many of my concerns.
>
> However, the revised paper does not adequately incorporate the suggested works for discussions [a, b] or experiments [c, d]. Additionally, I agree with other reviewers that the examination of variations across more random runs and the transfer of hyperparameters across datasets may require further attention. Importantly, the proposed experiments should be included in the camera-ready version or added to the supplemental material.
>
> In light of the above, I will maintain my current rating. However, I would be willing to increase my score if the authors adequately address my concerns in their revisions.

---

> > ### Author Response · Authors · 2024-11-26
> >
> > Thanks for your valuable comments.
> >
> >  **1. Examination of variations across more random runs.**
> >
> > Thanks for your suggestions. Following your suggestions, we conducted eight different runs on CIFAR-20 and STL-10, and cited the results from SCAN (10 runs), SeCu-Size (8 runs), and SeCu-Entropy (8 runs) from their original papers.
> >
> > The results in Table C-1 show that the variance of our method are significantly smaller than the comparison methods (1.3% on CIFAR-20 compared to SCAN's 2.7%, and 0.1% on STL-10 compared to SCAN's 1.9%). Our method not only achieves the lowest variance but also exhibits the best clustering performance, **benefiting from a prototype-based initialization strategy that enhances model robustness and training stability**.
> >
> > In Section C.2 of the revised paper, we have added the above experiments  and analyses. For the final version, we will extend all experiments to include results averaged over 8 runs with different seeds.
> >
> > **Table C-1: Results from multiple runs on CIFAR-20 and STL-10. We cited SCAN (10 runs), SeCu-Size (8 runs), and SeCu-Entropy (8 runs) from their original papers.**
> >
> > |                     |   CIFAR-20    |               |               |                 |    STL-10     |               |               |                 |
> > | ------------------- | :-----------: | :-----------: | :-----------: | :-------------: | :-----------: | :-----------: | :-----------: | :-------------: |
> > | Method              | ACC$\uparrow$ | NMI$\uparrow$ | ARI$\uparrow$ | ECE$\downarrow$ | ACC$\uparrow$ | NMI$\uparrow$ | ARI$\uparrow$ | ECE$\downarrow$ |
> > | SCAN-2              |   42.2±3.0    |   44.1±1.0    |   26.7±1.3    |        -        |   75.5±2.0    |   65.4±1.2    |   59.0±1.6    |        -        |
> > | SCAN-3              |   45.9±2.7    |   46.8±1.3    |   30.1±2.1    |        -        |   76.7±1.9    |   68.0±1.2    |   61.6±1.8    |        -        |
> > | SeCu-Size (mean)    |      50       |     50.7      |      35       |        -        |     80.2      |     69.4      |     63.9      |        -        |
> > | SeCu-Size (best)    |     51.6      |     51.6      |      36       |        -        |     81.4      |     70.7      |     65.7      |        -        |
> > | SeCu-Entropy (mean) |     49.9      |     50.6      |     34.1      |        -        |     79.5      |     68.7      |      63       |        -        |
> > | SeCu-Entropy (best) |     51.2      |     51.4      |     34.9      |        -        |     80.5      |     69.9      |     64.4      |        -        |
> > | CDC-Cal             | **59.4±1.3**  | **60.2±0.6**  | **44.9±1.1**  |   **3.5±1.2**   | **93.1±0.1**  |  **86±0.2**   | **85.6±0.3**  |   **1.1±0.2**   |
> >
> >
> >
> > **2. Transfer of hyperparameters across datasets.**
> >
> > Thanks for your suggestions. Following your suggestions, we **applied the hyperparameters from ImageNet10 (detailed in "Implementation Details" in Section 4 of the revised paper) to the 50, 100, and 200 subsets of ImageNet** and compared them with the SCAN method. We conducted the fair comparison under the same backbone network (ResNet-50) and the same subset divisions. For SCAN, we downloaded and tested the trained model from the SCAN code repository.
> >
> > **Results**. The results below demonstrate a robust improvement in calibration error metrics with our method, showing an average reduction of 8.4%. Meanwhile, the ACC of our method consistently surpasses that of the SCAN method.
> >
> > **Table C-2: Performance of SCAN and CDC on 50, 100, and 200 subsets of ImageNet.**
> >
> >   |         | ImageNet50 |         | ImageNet100 |          | ImageNet200 |          |
> >   | ------- | ---------- | ------- | ----------- | -------- | ----------- | -------- |
> >   |         | ACC        | ECE     | ACC         | ECE      | ACC         | ECE      |
> >   | SCAN-2  | 75.1       | 15.7    | 66.2        | 21.9     | 56.3        | 28.1     |
> >   | SCAN-3  | 76.8       | 14.2    | 68.9        | 18.8     | 58.1        | 25.8     |
> >   | CDC-Cal | **77.8**   | **7.5** | **71.2**    | **13.0** | **61.2**    | **13.2** |

---

> > ### Author Response · Authors · 2024-11-26
> >
> > **3. Incorporate the suggested works for discussions [a, b] or experiments [c, d]**.
> >
> > Thanks for your suggestions. The mentioned related works [a]\[b] have been already discussed in Line 116 and Lines 122-125 of the revised paper. The mentioned experiments [c]\[d] are also added in Appendix Section C.1 of the revised paper.
> >
> > **4. Revisions in the paper and the supplementary file**.
> >
> > The main revisions in the updated paper and supplementary file are summarized as follows:
> >
> > - **Related Work (Section 2):** We have refined and updated the discussion on calibration methods (Lines 122–127).
> >
> > - **Notation (Section 3):** We clarified the definitions of symbols and updated the corresponding formulas, Algorithm 1, and the illustration in Figure 2.
> >
> > - **Proposed Method (Section 3):** The order of method introduction has been adjusted to highlight dual-head framework and improve readability.
> >
> > - **Experiments (Section 4) in the main paper:** We streamlined the dataset descriptions and moved additional details to the appendix. An ablation study on the "Mini-cluster Number K" from Appendix B.5 of the original paper has been incorporated into the main body to provide a comprehensive evaluation.
> >
> > - **Implementation Details (Section B.1) in Appendix:** We have provided detailed settings for datasets and backbone networks, along with optimized descriptions of hyperparameters for comparative methods to enhance the readability and reproducibility of the paper.
> >
> > - **Ablation Study (Section B.5) in Appendix:** We have incorporated all the hyperparameter experiments mentioned in the comments (Batch Size \( B \), Number of Mini-cluster \( K \), Hidden Layer Size \( H \), and Loss Weight ($ w_{en}$ \) ), and provided detailed sensitivity analyses and insights into hyperparameter selection.
> >
> > - **Further Discussion (Section C) in Appendix:** We have included additional experiments based on the comments, including:
> >   - SSL experiments [c, d] (shown in **Section C.1**),
> >   - Examination of variability across more random runs (shown in **Section C.2**),
> >   - Transferability of hyperparameters across datasets (shown in **Section C.3**),
> >   - General applicability of CDC as a self-labeling stage (shown in **Section C.4**),
> >   - OOD detection for deep clustering methods (shown in **Section C.5**).
> >
> > The above revisions have been marked in red.
> >
> > If your concerns have been addressed, please help raise the score. If you have any other concerns, please let us know, and we will try our best to address them. Thanks.

---

> ### Comment · Reviewer_bpNw · 2024-11-27
>
> I appreciate the authors for conducting additional experiments that demonstrate the superiority of the proposed CDC. They have effectively addressed my concerns in their revisions. As a result, I will be increasing my score.

---

### Author Response · Authors · 2024-11-23

We thank all the reviewers for their valuable efforts in reviewing our paper and for providing insightful feedback.

We are delighted to see that they found our work: **novel** (`bpNw`, `yP9a`, `8jfQ`), **effective** (`bpNw`, `yP9a`, `8jfQ`, `KQtX`, `1qud`), and **theoretically sound** (`bpNw`, `8jfQ`, `1qud`). Furthermore, we also appreciate the reviewers highlighting that our work is extensively evaluated on diverse benchmarks, demonstrating **strong empirical results** (`bpNw`, `yP9a`, `8jfQ`, `1qud`), includes ablation studies that provide insights into the contributions of individual components (`yP9a`, `8jfQ`, `1qud`), and is **well-written** (`yP9a`, `8jfQ`) and **well-motivated** (`yP9a`, `8jfQ`, `1qud`).

Due to the **thorough theoretical and experimental validation**, the **pioneering insight in confidence calibration**, and the **universality of the self-labeling technique**, we believe this method can significantly influence numerous deep clustering methods in the future.

In the revised paper and responses to each reviewer, we provide additional insights to enhance the understanding of our approach. We welcome any additional questions or comments from the reviewers and will be glad to provide further clarifications or explanations as needed.

---

### Meta-Review · Area_Chair_a6Sa · 2024-12-13

**Metareview:**

This paper presents a work that addresses the over-confidence problem in deep clustering. A novel calibration method is proposed. A new network initialization strategy is also introduced to improve training efficiency. Both theoretical analysis and empirical evaluations are provided, which verify the superiority of the algorithm. The potential weakness of the work is the selection of hyperparameters. This paper provides useful insights for clustering. Reviewers and AC are overall satisfied with the rebuttal and revised manuscript. AC therefore recommends accepting this paper.

**Additional Comments On Reviewer Discussion:**

The following discussions and changes during the rebuttal are based on four reviewers. Reviewer KQtX is not involved here, because the comments and activities in the discussions are less informative. Specifically,
- Reviewer bpNw pointed out that the submission will benefit from a more comprehensive related work review. Besides, the details of algorithm flows should be presented better and explained clearly. The future work that targets the integration of previous insightful works can be discussed. After two rounds of responses, the raised concerns are addressed well, with an increased rating. This is also acknowledged by the reviewer.
- The concerns of reviewer yP9a are about experimental setups, unclear results, and the general applicability of the proposed method. The authors provided a series of supplementary experiments to answer the reviewer’s questions. The experimental results during rebuttal should be reflected in the final version of this paper, as mentioned.
- Reviewer 8jfQ commented on the hyperparameters, the stability of the proposed method, some misleading claims, and the confusion about the theory. After checking the rebuttal, except for the hyperparameter problem, the others have been properly solved. The hyperparameter issues are the main weakness of this work. Nevertheless, overall, AC considers that this paper still makes a solid contribution to the machine learning community, and can inspire future work a lot. Therefore, an acceptance recommendation is made.
- Reviewer 1qud was mainly concerned with the presentation of this paper and raised some questions about algorithm flows. The authors provide detailed descriptions accordingly. Although the rating of this reviewer is a bit negative, he/she claimed that he/she is satisfied with the revised manuscript and appreciated the detailed experiment results. Authors are expected to include detailed review questions and answers in the camera-ready version, which helps improve the readability of the work.

In summary, this paper provides valuable insights. Reviewers and AC acknowledge its contributions.

---

### Decision · Program_Chairs · 2025-01-22

Accept (Poster)